# An alternative splicing modulator decreases mutant HTT and improves the molecular fingerprint in Huntington's disease patient neurons

Huntington's disease (HD) is a neurodegenerative disorder caused by poly-Q expansion in the Huntingtin (HTT) protein. Here, we delineate elevated mutant HTT (mHTT) levels in patient-derived cells including fibroblasts and iPSC derived cortical neurons using mesoscale discovery (MSD) HTT assays. HD patients' fibroblasts and cortical neurons recapitulate aberrant alternative splicing as a molecular fingerprint of HD. Branaplam is a splicing modulator currently tested in a phase II study in HD (NCT05111249). The drug lowers total HTT (tHTT) and mHTT levels in fibroblasts, iPSC, cortical progenitors, and neurons in a dose dependent manner at an $IC_{50}$ consistently below 10 nM without inducing cellular toxicity. Branaplam promotes inclusion of non-annotated novel exons. Among these Branaplam-induced exons, there is a 115 bp frameshift-inducing exon in the HTT transcript. This exon is observed upon Branaplam treatment in Ctrl and HD patients leading to a profound reduction of HTT RNA and protein levels. Importantly, Branaplam ameliorates aberrant alternative splicing in HD patients' fibroblasts and cortical neurons. These findings highlight the applicability of splicing modulators in the treatment of CAG repeat disorders and decipher their molecular effects associated with the pharmacokinetic and -dynamic properties in patient-derived cellular models.

Huntington's disease (HD) is a progressive neurodegenerative disorder caused by CAG-repeat expansion in the coding region of the Huntingtin (HTT) transcript, leading to an elongated polyglutamine (poly-Q) stretch in HTT[1]. While individuals with 40 and above CAG-repeats in one allele will develop HD, a repeat length below 36 is non-pathogenic. Intermediate repeat lengths in the range between 36 and 39 may cause HD with incomplete penetrance[2]. The elongated poly-Q, mutant HTT (mHTT) is suspected to exhibit a toxic gain of function resulting in neuronal toxicity[3].

Clinically, HD is characterized by the triad of motor dysfunction, psychiatric symptoms, and cognitive deficits. Motor symptoms, most prominently chorea, are related to the degeneration of striatal medium spiny neurons[2] and subsequently cortical projection neurons. Interestingly, cortico-striatal projections are affected years prior to disease onset[4]. Specifically, degeneration of pyramidal neurons in the primary motor cortex and anterior cingulate cortex lead to more pronounced motor and mood impairments, respectively[5,6].

So far, there is no causal therapy to improve or even halt the disease course of HD. However, a number of recent clinical trials focus on small compounds lowering HTT levels[7]. One of the latest clinical trials investigates the effectiveness of the alternative splicing (AS) modulator Branaplam (previously called LMI070, NVS-SM1) in HD

✉ e-mail: beate.winner@fau.de; juergen.winkler@uk-erlangen.de

(NCT05111249). Branaplam was initially designed for spinal muscular atrophy (SMA) and promotes inclusion of exon 7 in the SMN2 transcript via stabilization of the pre-mRNA - U1 snRNP complex[8].

Here, we describe Branaplam's mechanism of action and effects in HD. We develop a HD patient-derived cellular platform to investigate mHTT levels using a validated HTT assay. We demonstrate that Branaplam is able to reduce mHTT levels. Furthermore, Branaplam restores mHTT-induced aberrant splicing, an important molecular feature of HD, in transcripts that are not directly targeted by Branaplam. We evaluate the precise pharmacokinetic and -dynamic properties of Branaplam in primary fibroblasts and iPSC-derived cortical neurons of Ctrl and HD patients. Lastly, we explore the small

molecule's potential to revert molecular phenotypes of HD patients' neurons.

## Results

### mHTT is increased in cellular model of HD patients

We chose to use various patient-derived non-neuronal and neuronal cells to model HD in vitro (Fig. 1a). Besides four Ctrls, we recruited four HD patients originating from three distinct families (Table 1 and Supplementary Fig. 1a). The CAG repeats on the affected allele ranged from 39 to 57 (Table 1). Clinically, all patients presented motor symptoms. Hence, we reprogrammed fibroblasts into iPSC and subsequently differentiated them into cortical neurons using a previously

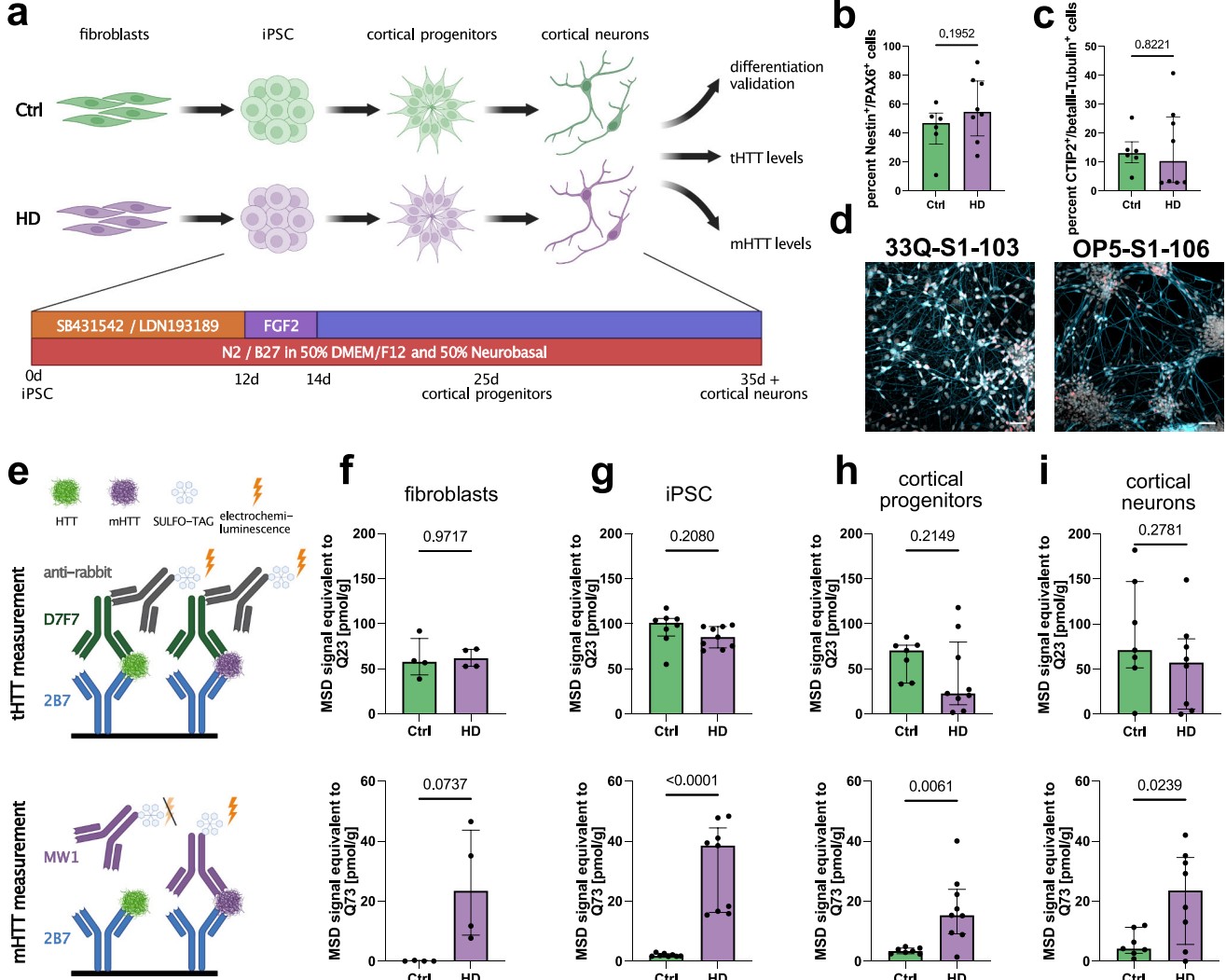

**Fig. 1 | Mutant HTT is increased in HD patient-derived cells using an MSD assay.** **a** Paradigm illustrating the HD patient-based disease model (fibroblasts, iPSC, cortical progenitors (25d old), and cortical neurons (35d old)) and readouts. Created with BioRender.com. **b** Bar plot depicting FACS quantification of NESTIN/ PAX6 double-positive cells. Statistics: Welch's test. Bars: median ± IQR. **c** Bar plot illustrating FACS quantification of bIII-Tubulin/CTIP2 double-positive cells. Statistics: Welch's test. Bars: median ± IQR. **d** Representative pictures of cortical neurons. Scale bar 50 μm. **e** Illustration depicting the MSD HTT quantification assay, where the added protein samples bind to 2B7 antibody, used for coating the plates. The SULFO-TAG coupled antibodies D7F7 and MW1 are added for quantification of total HTT and mutant HTT, respectively. Note: numeric values from 2B7/D7F7 assay (total HTT) cannot be directly set in relation to numeric values from 2B7/MW1 assay (mutant HTT). Created with BioRender.com. **f** Bar plots quantifying total (tHTT, top) and mutant (mHTT, bottom) levels in

fibroblasts (4 Ctrl lines, 4 HD lines) with 2B7/D7F7 and 2B7/MW1 MSD assays, respectively. Statistics: tHTT: Welch's test (P value = 0.9717); mHTT Welch's test (P value = 0.0737). Bars: median ± IQR. **g** Bar plots quantifying total (tHTT, top) and mutant (mHTT, bottom) levels in iPSC (8 Ctrl lines, 9 HD lines) with 2B7/D7F7 and 2B7/MW1 MSD assays, respectively. Statistics: tHTT: Welch's test (P value = 0.2080); mHTT Mann–Whitney test (P value < 0.0001). Bars: median ± IQR. **h** Bar plots quantifying total (tHTT, top) and mutant (mHTT, bottom) levels in cortical progenitors (7 Ctrl lines, 9 HD lines) with 2B7/D7F7 and 2B7/MW1 MSD assays, respectively. Statistics: tHTT: Welch's test (P value = 0.2149); mHTT Welch's test (P value = 0.0061). Bars: median ± IQR. **i** Bar plots quantifying total (tHTT, top) and mutant (mHTT, bottom) levels in cortical neurons (7 Ctrl lines, 8 HD lines) with 2B7/D7F7 and 2B7/MW1 MSD assays, respectively. Statistics: tHTT: Welch's test (P value = 0.2781); mHTT Welch's test (P value = 0.0239). Bars: median ± IQR. Source data are provided as a Source Data file.

**Table 1 | HD and Ctrl cohort**

| donor | gender | Age at biopsy | CAG repeats | fibroblast ID | iPSC ID |
|---|---|---|---|---|---|
| HD | f | 44 | 16/40 | UKERf4Q4 | UKERi4Q4-S1-105<br>UKERi4Q4-S1-109[a] |
| HD | m | 54 | 15/39 | UKERfOP5 | UKERiOP5-S1-106[a]<br>UKERiOP5-S1-108 |
| HD | m | 26 | 18/50 | UKERf59H | UKERi59H-S1-101<br>UKERi59H-S1-103[a]<br>UKERi59H-S1-108 |
| HD | f | 23 | 28/57 | UKERf919 | UKERi919-S1-101[a] |
| Ctrl | f | 45 | 15/16 | UKERf33Q | UKERi33Q-S1-109[a] |
| Ctrl | m | 43 | 17/21 | UKERfB26 | UKERiB26-S1-007<br>UKERiB26-S1-018[a] |
| Ctrl | m | 52 | 18/19 | UKERf4CC | UKERi4CC-S1-007<br>UKERi4CC-S1-015[a] |
| Ctrl | m | 32 | 16/17 | UKERf4L6 | UKERi4L6-S1-027[a]<br>UKERi4L6-S1-032 |

*HD* Huntington's disease, *Ctrl* control.
[a]Marks iPSC line setup used for all experiments starting Fig. 2. Underlined parts of IDs indicate abbreviated form of line name used in figures.

published dual-SMAD-inhibition-based protocol[9]. At day 25 of differentiation, NESTIN⁺/PAX6⁺ cortical progenitors were observed (Fig. 1b and Supplementary Fig. 1b, c), and from day 35 on deep layer cortical neurons positive for CTIP2 were apparent (Fig. 1c, d and Supplementary Fig. 1d, e). No differences in differentiation potential between Ctrl and HD were detected (Fig. 1b, c).

Next, we determined the total (tHTT) and mHTT levels in cellular homogenates. We used a mesoscale discovery (MSD) assay with an electrochemiluminescent readout. MSD assays are applied to semi-quantitatively determine changes in protein levels. An N-terminally-binding HTT antibody (2B7) captures HTT and an antibody binding a central part of HTT downstream of the poly-Q tract (D7F7) or an antibody with preferred binding to elongated poly-Q (MW1) is used to quantify tHTT (2B7/D7F7) and mHTT (2B7/MW1) via a SULFO-TAG, respectively. The resultant HTT signal values are back-calculated to a standard of recombinant HTT with 23 (Q23) and 73 (Q73) glutamines (Fig. 1e). Importantly, the obtained signal values of the tHTT assay (2B7/D7F7) and mHTT assay (2B7/MW1) in a given sample cannot be directly set into relation with each other (e.g., no mHTT/tHTT ratio calculations possible) due to different properties of both assays (explained in detail in "Methods")[10]. While tHTT levels were unchanged in all analyzed cell types, a substantial increase in signal in the mHTT assay was observed in HD-patient iPSC, cortical progenitors, and cortical neurons (Fig. 1f–i). Hence, our cellular platform is suitable to detect changes in mHTT levels in non-neuronal and neuronal cells.

### HD patient-derived cells exhibit aberrant RNA missplicing as a molecular disease phenotype

Aberrant AS events are present in patients' primary motor cortex[11] and striatum[12]. Therefore, we investigated whether HD-patients' cells also exhibit similar molecular fingerprints by performing deep long read RNA-sequencing of HD and Ctrl fibroblasts (four patients vs. four healthy controls) and iPSC-derived cortical neurons (3 patients vs. 3 healthy controls) (Fig. 2a). We identified 306 and 1437 significantly (FDR < 0.05; inclusion level difference > 0.1) differentially spliced exons in fibroblasts (Fig. 2b) and iPSC-derived cortical neurons of HD patients (Fig. 2c), respectively, consisting mostly of cassette exons (SE) (Fig. 2d, e). Eleven included and excluded events were shared between HD fibroblasts and HD cortical neurons (Fig. 2f). Next, we asked which upstream AS factors may regulate these changes. Therefore, we made use of the ENCODE-enhanced cross-linking and immunoprecipitation (eCLIP) sequencing experiments, providing RNA-binding profiles of a large number of RNA-binding proteins (RBPs)[13]. We investigated

whether binding of these proteins to the RNA of HD AS sites encompassing the region from the upstream exon start site to the downstream exon end was enriched over background. HD-AS events in fibroblasts and iPSC-cortical neurons were enriched for distinct RBPs (Fig. 2g, h). Interestingly, HD-AS events in iPSC-cortical neurons were enriched for binding sites of RBFOX2, TIA1, and U2AF2 (Fig. 2h). Importantly, identical RBPs were previously implicated in aberrant AS in HD postmortem tissue[12]. Next, we asked whether the enriched RBPs exhibited a change in gene expression. Elevated levels of HLTF and a reduction of PCBP2 were found in fibroblasts, whereas a reduction of U2AF1 and U2AF2 was detected in cortical neurons of HD patients (Supplementary Fig. 2a, b). A reduction of U2AF2 was previously reported in HD[12]. Protein insolubility is an important feature in neurodegenerative diseases and we have recently shown that changed biochemical solubility properties of certain RBPs are associated with aberrant AS in amyotrophic lateral sclerosis (ALS)[14]. We investigated the soluble and insoluble protein fractions of Ctrl and HD cortical neurons of five candidate RBPs that are enriched in cortical neuron HD AS events (ILF3, QKI, U2AF2, RBFOX2, and TIAL1) (Supplementary Fig. 3a–f). ILF3, U2AF2, RBFOX2, and TIAL1 did not exhibit significant changes in the level of solubility. A modest but significant increase in soluble QKI was observed in HD cortical neurons (Supplementary Fig. 3c). This suggests that aberrant RNA missplicing is present in HD patients' fibroblasts and to a larger extent in iPSC-derived cortical neurons. These changes may be due to altered gene expression and biochemical properties of RBPs. In conclusion, HD patients' iPSC cortical neurons recapitulate aberrant AS as a major molecular pattern of HD.

### Branaplam/LMI070 reduces total and mutant HTT protein levels in HD patients' cells

Therapeutic strategies in HD aim at lowering HTT levels to reduce mHTT toxicity. A recently initiated phase II study repurposes the small molecule AS modulator Branaplam (LMI070, NCT05111249). Hence, we sought to investigate its effects to specifically reduce mHTT levels and its impact on Ctrl and HD-derived cells (Fig. 3a). Branaplam exhibited dose-dependent effects in lowering HTT levels in fibroblasts and iPSC (Fig. 3b, c, tHTT). Similar trends were visible for mHTT in HD patients' cells (Fig. 3b, c, mHTT). Interestingly, Branaplam treatment did not induce cellular toxicity in fibroblasts and iPSC (Fig. 3b, c, toxicity). In order to investigate the pharmacokinetic properties of Branaplam in vitro, we performed a dose−response experiment in human cortical progenitor cells. The half-maximal inhibitory concentration (IC50) of Branaplam was consistently below 10 nM, reducing

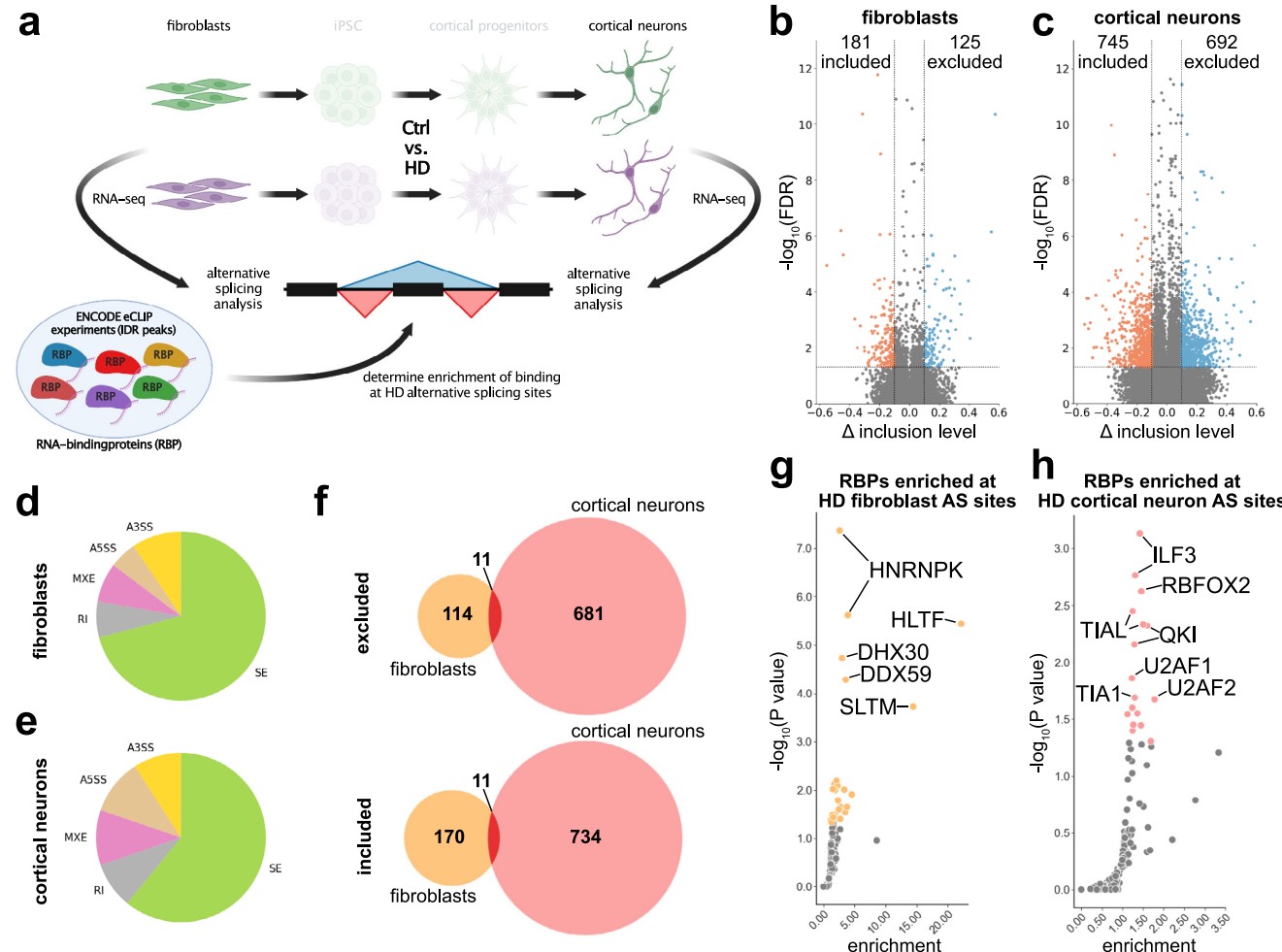

**Fig. 2 | Aberrant alternative splicing is present in HD patients' fibroblasts and cortical neurons. a** Paradigm illustrating the analysis of fibroblasts and iPSC-derived cortical neurons (50d old) and integration of publicly available RNA-binding profile data (ENCODE eCLIP-seq) to determine degree and origin of alternative splicing in HD. Created with BioRender.com. **b** Volcano plot showing inclusion levels difference (x) and significance (y) of alternative splicing events in Ctrl-DMSO vs HD-DMSO fibroblasts. Red: significantly included splicing events; blue: significantly excluded splicing events. Horizontal dashed line: FDR = 0.05; vertical dashed lines at −0.1 and 0.1. **c** Volcano plot showing inclusion levels difference (x) and significance (y) of alternative splicing events in Ctrl-DMSO vs HD-DMSO iPSC-derived cortical neurons. Red: significantly included splicing events; blue: significantly excluded splicing events. Horizontal dashed line: FDR = 0.05; vertical dashed lines at −0.1 and 0.1. **d** Pie chart of alternative splicing types in significant HD alternative splicing events in fibroblasts. Green: cassette exons (SE); yellow: alternative 3' splice site (A3SS);

brown: alternative 5' splice site (A5SS); pink: mutually exclusive exons (MXE); gray: retained introns (RI). **e** Pie chart of alternative splicing types in significant HD alternative splicing events in iPSC-derived cortical neurons. Green: cassette exons (SE); yellow: alternative 3' splice site (A3SS); brown: alternative 5' splice site (A5SS); pink: mutually exclusive exons (MXE); gray: retained introns (RI). **f** Venn diagram showing overlap of significantly differentially spliced events in HD in fibroblasts and in neurons, respectively. Red depicts overlap between both cell types. Top graph: exon excluded in HD; bottom graph: exon included in HD. **g** Scatter plot illustrating RNA-binding protein (RBP) RNA-binding enrichment (x) and significance (y) at HD alternative splicing events in fibroblasts. Yellow colored dots depict RBPs with P value ≤ 0.05. **h** Scatter plot illustrating RNA-binding protein (RBP) RNA-binding enrichment (x) and significance (y) at HD alternative splicing events in iPSC cortical neurons. Salmon-colored dots depict RBPs with P value ≤ 0.05. Source data are provided as a Source Data file.

total as well as mHTT levels without affecting cellular toxicity (Fig. 3d). A concentration of 10 nM reduced tHTT levels in iPSC-derived cortical neurons by 38.8% and mHTT levels in HD patients by 21.8% without inducing toxicity (Fig. 3e). To determine potential toxic effects of Branaplam on neuronal subtypes, we investigated Caspase-3/7 activation in deep layer CTIP2-positive neurons. Branaplam did not induce cell death in CTIP2⁺ and CTIP2⁻ neurons (Fig. 3f and Supplementary Fig. 4a–c). In addition, we explored the impact of Branaplam on the proliferation of SOX2⁺ neural progenitor cells via EdU incorporation assay. No changes were observed in proliferation upon 3 days Branaplam treatment (Supplementary Fig. 4d, e). In summary, these findings suggest that Branaplam efficiently reduces total and mutant HTT protein levels in various Ctrl and HD patient-derived cell types without inducing toxicity and altering proliferation.

## Branaplam promotes inclusion of previously non-annotated novel splice sites

Next, we explored how the splicing modulator Branaplam leads to a reduction in HTT protein levels. We performed a streamlined AS analysis in fibroblasts and cortical neurons of controls and HD patients with and without Branaplam treatment to decipher Branaplam's targets, sequence preferences, and effects on gene expression in an unsupervised manner (Fig. 4a). Significantly differentially spliced events upon Branaplam treatment in all four cohorts (Ctrl fibroblasts, HD fibroblasts, Ctrl cortical neurons, HD cortical neurons) were grouped using k-means, resulting in 10 distinct clusters (cluster 0–cluster 9) (Fig. 4b). Cluster 6 and cluster 9 exhibited coherent, unidirectional alternative splicing changes (exon inclusion) in all four cohorts (Fig. 4c, d and Supplementary Fig. 5a). Interestingly, more than

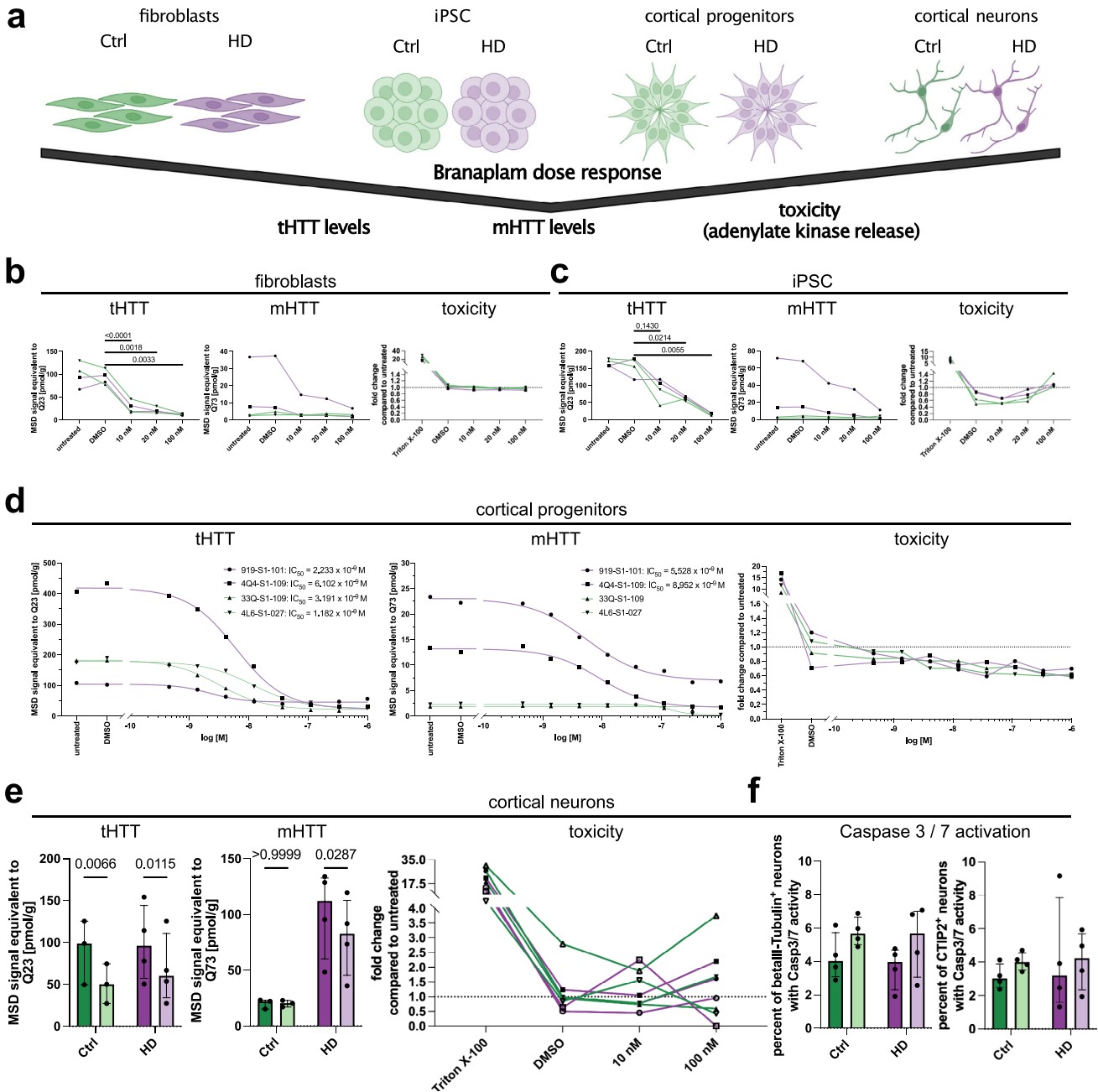

**Fig. 3 | Branaplam reduces total and mutant HTT protein levels in a dose-dependent manner without inducing toxicity. a** Paradigm: cell types and parameters analyzed. Created with BioRender.com. **b**–**d** tHTT levels (2B7/D7F7 assay), mHTT levels (2B7/MW1 assay) and toxicity (adenylate kinase release, Triton: positive control) in fibroblasts (**b**), iPSC (**c**) and cortical progenitors (**d**) with different Branaplam concentrations for 72 h. Green: Ctrl samples ($n = 2$ 33Q/33Q-S1-109, 4L6/4L6-S1-027), purple: HD samples ($n = 2$ 919/919-S1-101, 4Q4/4Q4-S1-109). **b** Fibroblasts statistics: tHTT: one-way ANOVA with Geisser−Greenhouse correction ($P = 0.0012$); mHTT: no statistics applied; toxicity (Triton excluded): Friedman test ($P = 0.0770$). **c** iPSC statistics: tHTT one-way ANOVA with Geisser−Greenhouse correction ($P = 0.0032$); mHTT: no statistics applied; toxicity (Triton excluded): one-way ANOVA with Geisser−Greenhouse correction ($P = 0.0455$), no significant differences in multiple comparisons. **d** cortical progenitor statistics: tHTT IC$_{50}$: 919-S1-101 = $2.233 \times 10^{-9}$ M; 4Q4-S1-109 = $6.102 \times 10^{-9}$ M; 33Q-S1-109 = $3.191 \times 10^{-9}$ M; 4L6-S1-027 = $1.182 \times 10^{-9}$ M; mHTT: 919-S1-101 = $5.528 \times 10^{-9}$ M; 4Q4-S1-109 = $8.952 \times 10^{-9}$ M;

33Q-S1-109 and 4L6-S1-027 = not calculated; toxicity (Triton excluded): one-way ANOVA with Geisser−Greenhouse correction ($P = 0.06$). **e** tHTT levels (2B7/D7F7 assay), mHTT levels (2B7/MW1 assay) (Ctrl $n = 3$; HD $n = 4$) and toxicity (adenylate kinase release, Triton as positive control) in cortical neurons of (Ctrl $n = 4$; HD $n = 4$). tHTT and mHTT measured with DMSO (dark shades) or 10 nM Branaplam (light shades) for 72 h. Statistics: tHTT: 2-way ANOVA (DMSO vs. Branaplam: $P = 0.0009$; Ctrl vs. HD: $P = 0.6820$; interaction: $P = 0.3833$); mHTT: two-way ANOVA (DMSO vs. Branaplam: $P = 0.0614$; Ctrl vs. HD: $P = 0.0223$; interaction: $P = 0.0615$); toxicity (Triton excluded): Friedman test ($P = 0.5306$). Bars: median ± IQR. **f** Bar plot showing number of Casp3/7 positive beta-III-Tubulin+ and CTIP2+ cortical neurons after DMSO (dark shades) or (light shades) 72 h 10 nM Branaplam treatment (Ctrl $n = 4$; HD $n = 4$). Statistics: beta-III-Tubulin+: two-way ANOVA (DMSO vs. Branaplam: $P = 0.0782$; Ctrl vs. HD: $P = 0.4744$; interaction: $P = 0.9502$); CTIP2+: 2-way ANOVA (DMSO vs. Branaplam: $P = 0.7230$; Ctrl vs. HD: $P = 0.5779$; interaction: $P = 0.6348$). Bars: median ± IQR. Source data are provided as a Source Data file.

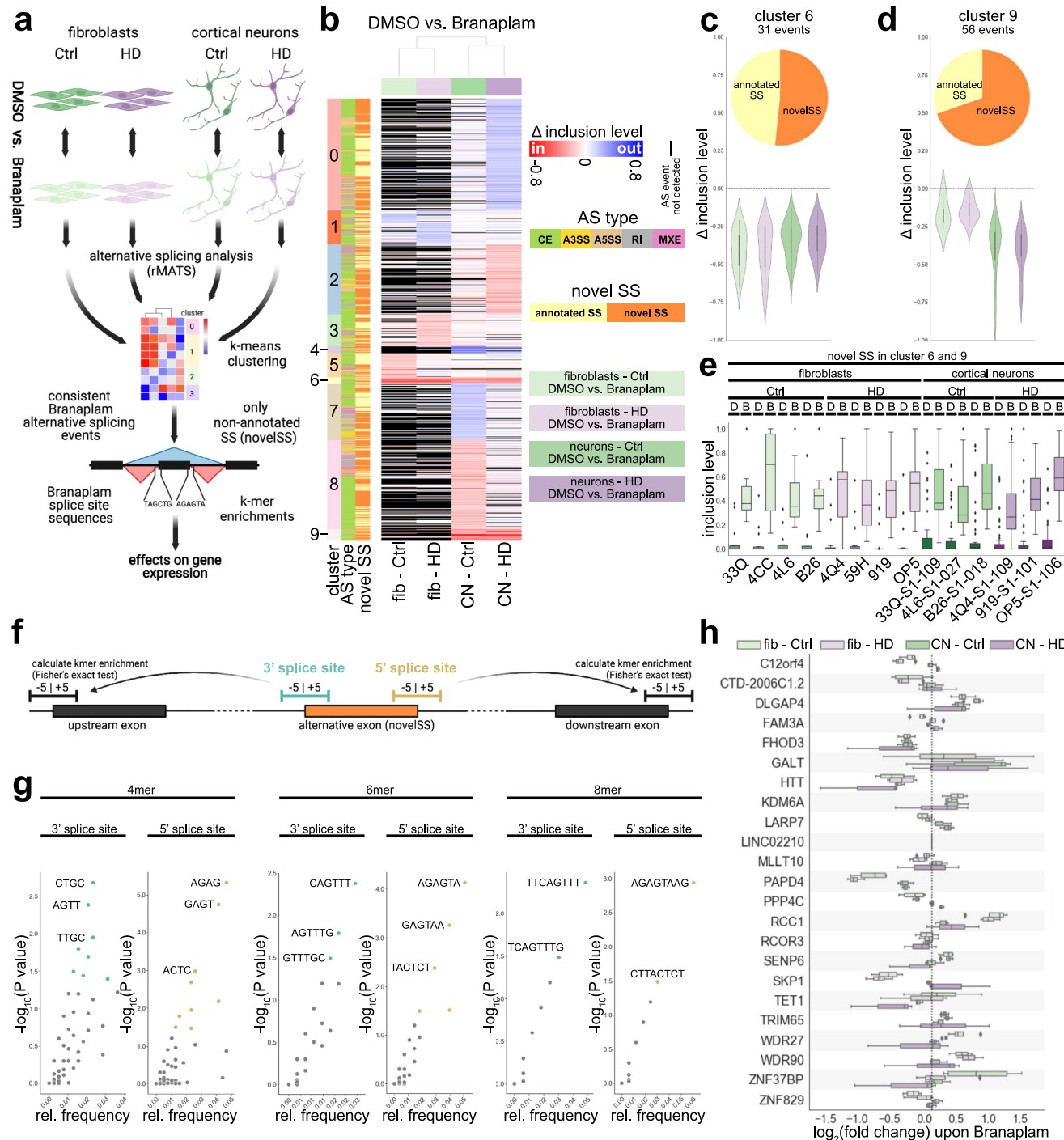

**Fig. 4 | Branaplam promotes non-annotated exon inclusion at preferred sequences. a** Paradigm illustrating analysis pipeline in the four analyzed conditions (light green: fibroblasts Ctrl DMSO vs. 10 nM Branaplam (4 vs. 4); light purple: fibroblasts HD DMSO vs. 10 nM Branaplam (4 vs. 4); green: cortical neurons (50d) Ctrl DMSO vs. 10 nM Branaplam (3 vs. 3); purple: cortical neurons (50d) HD DMSO vs. 10 nM Branaplam (3 vs. 3)). rMATS for AS detection and quantification, k-means clustering to identify consistently changed AS events, kmer enrichment to investigate sequence preferences. Created with BioRender.com. **b** Heatmap illustrating inclusion level differences of alternative splicing events significant (FDR ≤ 0.05; absolute inclusion level difference ≥0.1) in at least one of the four comparisons from included (red colors) to excluded (blue colors). Events not detected in an analysis are marked black. Y axis colors show AS cluster (0–9, determined by k-means), AS type and novel splice sites. **c, d** Violin plot and pie chart of AS events in cluster 6 (**c**) and cluster 9 (**d**). Violin plot shows inclusion level difference in the four comparisons (Ctrl: greens; HD: purples; fibroblasts: light shades; cortical neurons: dark shades). Pie chart illustrates distribution of annotated (yellow) vs. non-annotated exons (novelSS, orange). **e** Box plot showing inclusion values of AS events in cluster 6 and 9 that are non-annotated (novelSS) in individual samples. D: DMSO, B: Branaplam. **f** Paradigm illustrating strategy to determine sequence preferences of Branaplam-induced AS at the 5′ exon (cyan) and 3′ exon end (gold). **g** Scatter plots of 4mer, 6mer and 8mer relative frequencies (x) and significance (y) at 3′ splice site (respective left graph) and 5′ splice site (respective right graph). Colored dots represent significant enrichments. **h** Box plot illustrating gene expression changes (log₂(RPKM Branaplam/ RPKM DMSO)) of genes with AS events in cluster 6 and 9 that are non-annotated (novelSS) in all four comparisons. fib fibroblasts, CN cortical neurons. Source data are provided as a Source Data file.

50% of events in both clusters were novel splice sites (novelSS) that represent previously non-annotated exons (Fig. 4c, d, pie charts). In contrast to the annotated exons, the 55 novelSS exons are predominantly excluded in untreated cells and only become apparent after Branaplam treatment (Fig. 4e and Supplementary Fig. 5b). We compared our Branaplam-induced exons (clusters 6 and 9) to Branaplam-induced exons that have been previously characterized in HEK293 cells[15]. Interestingly, out of 25 events discovered by Monteys and colleagues to be exclusively regulated by Branaplam, 15 were also present within the novelSS exons induced by Branaplam (Supplementary Fig. 5c). This suggests a very high validity and robustness of the present analyses and therefore the identified events. Next, we analyzed if those novel, Branaplam-dependent exons exhibited enrichment of specific sequences around their 5' and 3' splice site by analyzing 4mer, 6mer and 8mer sequences (Fig. 4f, g). Interestingly, even at the 8mer level we identify enriched sequences at the 3' splice site (TTCAGTTT) and 5' splice site (AGAGTAAG) (Fig. 4g), suggesting, at least in part, a sequence-dependent mode of action of Branaplam. NovelSS exons may contain STOP codons or result in an out-of-frame transcript potentially leading to nonsense-mediated RNA decay and mRNA degradation. Therefore, we analyzed gene expression changes of the transcripts that contain the 55 newly identified Branaplam-induced exons. Three transcripts (DLGAP4, RCC1, and KDM6A) exhibited a consistent increase in gene expression (Fig. 4h). Interestingly, we identified that the levels of FHOD3, PAPD4 and also HTT were consistently reduced in all four comparisons (Fig. 4h).

We further evaluated the HTT transcript and detected an inclusion of a 115b long frameshift-inducing exon with 2 STOP codons between exon 49 and exon 50 upon Branaplam treatment (Fig. 5a–c). This splice site contained the previously identified Branaplam associated 5' splice site sequence AGAGTAAG (Fig. 5a). We further validated the integration of this exon by RT-PCR with primers annealing to the flanking exons (Fig. 5a, blue arrows). A consistent integration of this exon in all analyzed Ctrl and HD patient fibroblasts and cortical neurons was observed (Fig. 5d–g), leading to reduced HTT mRNA levels (Fig. 5h, i). In summary, these findings suggest that Branaplam promotes novelSS exon inclusion with distinct sequence preferences. This is present in HTT transcripts leading to a nonsense-mediated RNA decay isoform and a profound reduction of HTT levels.

### Aberrant alternative splicing pathology in HD is ameliorated by Branaplam

We have identified aberrant AS in HD fibroblasts and iPSC-derived cortical neurons as a molecular HD fingerprint (Fig. 2). Next, we investigated if Branaplam treatment, and the accompanied reduction in mutant HTT levels, improves the AS deficiency in HD (Fig. 6a). Branaplam significantly reduced the absolute inclusion level differences of HD AS events by 27.6% in fibroblasts and 28.6% in iPSC-derived cortical neurons (Fig. 6b, c). In total, 53.2% of HD AS events in fibroblasts (Fig. 6d–f) and 47.9% of HD AS events in iPSC-derived cortical neurons (Fig. 6g–i) exhibited an absolute inclusion level difference below our threshold of 0.1 upon Branaplam treatment. This suggests that Branaplam ameliorates a prominent molecular signature in HD iPSC-derived cortical neurons. Branaplam itself did not directly target the HD-specific AS events (Supplementary Fig. 6a, b). The rescued AS events were preferentially bound by certain RBPs (Supplementary Fig. 6c, d), e.g., rescued events in cortical neurons were more prominently bound by QKI than non-rescued events (Supplementary Fig. 6d). This suggests that Branaplam may revert HD AS events indirectly by reduction of mHTT RNA and protein levels consequently altering RBPs' functions.

## Discussion

This study describes the reduction of mHTT levels in HD patients' fibroblasts and iPSC-derived cortical neurons by application of the splicing modulator Branaplam without inducing cellular toxicity. Specifically, we show that aberrant AS is ameliorated following Branaplam treatment.

Various approaches that were and still are under clinical development for the treatment of HD focus on lowering HTT levels. This includes antisense oligonucleotides (ASOs)[16] (Generation HD 1: NCT03761849; Precision HD-1: NCT03225833; Precision HD-2: NCT03225846) and adeno-associated virus (AAV)-mediated gene therapeutic delivery of RNAi-based machineries (NCT04120493), currently both on hold. Both approaches are dependent on repeated intrathecal or stereotactic injections. In contrast, Branaplam is an orally available small molecule, much easier applicable in HD patients.

Here, we delineated the mechanism of action of Branaplam in two distinct cell types in Ctrl and HD patients. Branaplam was originally designed to promote inclusion of exon 7 in the SMN2 transcript as an intervention for SMA[8]. We reveal that Branaplam also induces inclusion of multiple non-annotated novel exons, preferentially exons with AGAGTAAG sequences at their 5' splice site. Among these, there is a frameshift-inducing exon in the HTT transcript, leading to a profound lowering of tHTT and mHTT levels in Ctrl and HD patient cells. A recent study identifies a similar mechanism of action of Branaplam in a permanent neuroblastoma cell line of human origin[17]. Furthermore, we confirmed the mechanism of action in multiple Ctrl and HD patient cell types, including iPSC-derived cortical neurons. Additionally, we precisely defined the pharmacokinetic properties of the small molecule AS modifier using validated quantitative assays for of tHTT and mHTT that showed an $IC_{50}$ consistently below 10 nM in Ctrl and HD patient cells. We did not observe toxic effects of Branaplam in vitro. This included no change human cortical progenitor proliferation, confirming previous studies that investigated proliferation in the subventricular zone of dogs and rats upon Branaplam administration[18]. Furthermore, the efficacy of Branaplam for HD was recently demonstrated by phenotype improvements in a HD mouse model upon administration[17]. We further underscore the effectiveness of this molecule by providing compelling evidence that Branaplam ameliorates a molecular fingerprint in a human HD in vitro model. However, it is of speculative nature if rescue of these AS events contributes to the clinical efficacy of the drug. On a broader scope, we emphasize the applicability of AS modulators to alter pathological protein levels by integration of non-annotated exons and restore molecular fingerprints using primary fibroblasts of HD patients and patient iPSC-derived cortical neurons.

Our iPSC-based model recapitulates aberrant AS as a feature of HD that has been previously observed in postmortem tissue[11,12]. Six proposed candidate AS events in postmortem tissue (within the transcripts of CCDC88C, KCTD17, SYNJ1, VPS13C, TRPM7, SLC9A5) are not recapitulated in our iPSC-based neuronal dataset. However, there appears to be a similarity of the RBPs driving aberrant AS in the present iPSC neuronal and previously published postmortem tissue[12], suggesting the possibility of shared changes in RBP function and RNA processing in both systems. Aberrant AS is an interesting phenotype in the context of neurodegenerative diseases and has been most frequently studied in ALS patients. Widespread AS changes are observed in postmortem tissue[19,20]. This phenotype was also observed in ALS iPSC models[14,21] that are driven by biochemical and cellular alterations in specific RBPs[14]. Our findings give only a glimpse into AS in HD, but show that iPSC cortical neurons may be a powerful model to study this distinct HD-associated phenotype. However, there is an urgent need for future studies thoroughly dissecting the origin of aberrant AS in HD.

## Methods

### Subjects and human samples

The generation and use of human iPSCs were approved by the Institutional Review Board (Nr. 4120 and 259_17B: *Generierung von*

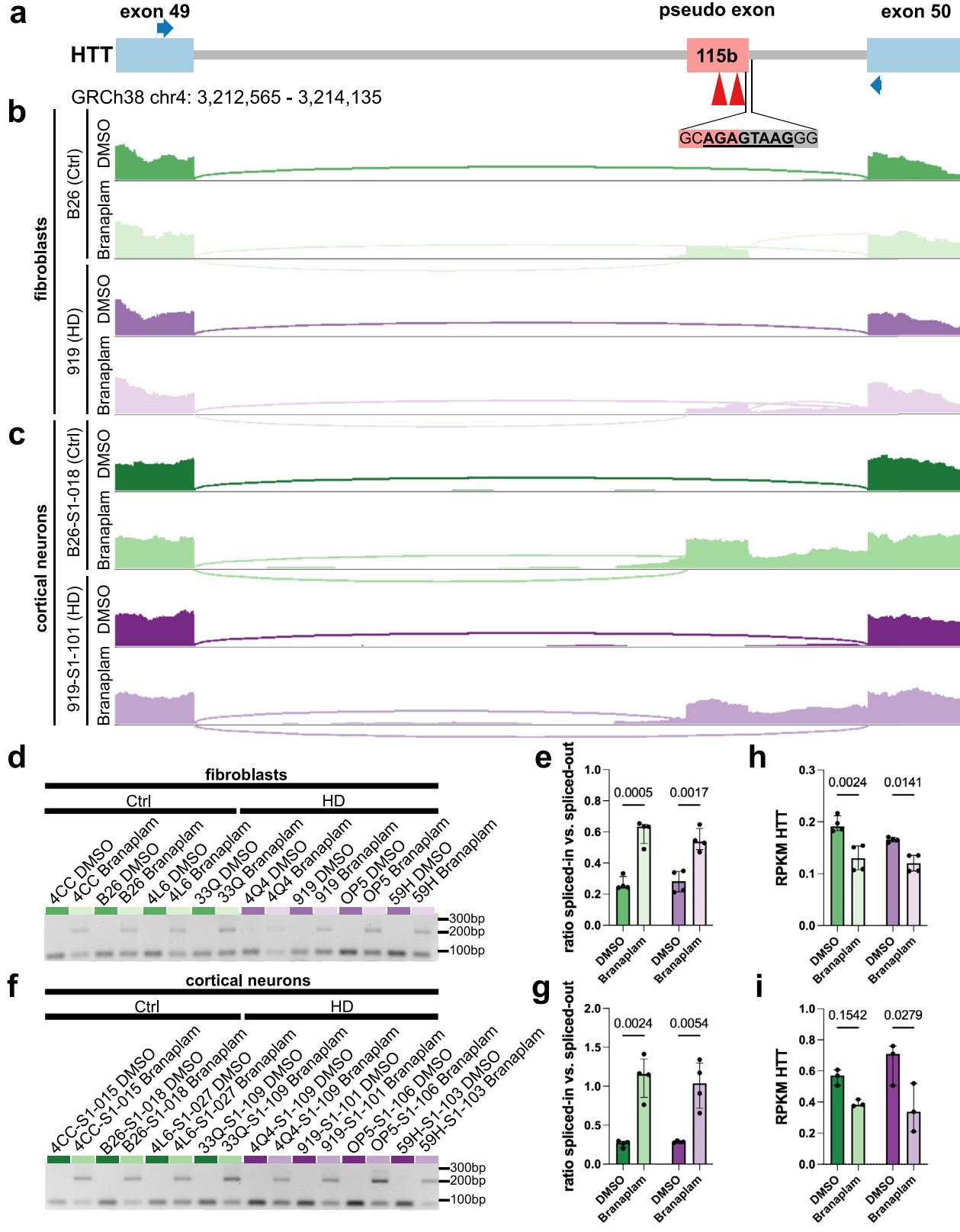

humanen neuronalen Modellen bei neurodegenerativen Erkrankun-
gen*). Formal informed consent was obtained from all subjects. Four
patients from three different families and age-matched controls
without history of neurological disorders were recruited. CAG
repeats of fibroblasts and iPSCs were measured by the center of
Human Genetics at the University Hospital Regensburg (Ute
Hehr, MD).

**Fibroblast culture**

Fibroblasts were resuspended in fibroblast growth medium (FGM, 75%
DMEM, 15% FCS, 2 mM L-glutamine, 100 µg/ml penicillin/streptomycin,
2 ng/ml fibroblast growth factor 2) and plated on polystyrene cell
culture flasks. Medium was changed twice a week. Fibroblasts were
split by removing FGM, adding Trypsin supplemented with 0.05%
ethylenediaminetetraacetic acid (EDTA), and incubating at 37 °C until

**Fig. 5 | A frameshift-inducing exon is inserted in HTT upon Branaplam treatment leading to a reduction of HTT mRNA levels. a** Illustration of locus with novel exon integrated in HTT transcript. Red arrowheads: in-frame STOP codons, blue arrows: primer locations for PCR validation. **b** Sashimi plot illustrating read density and junction-spanning reads in Ctrl and HD fibroblasts with and without Branaplam. **c** Sashimi plot illustrating read density and junction-spanning reads in Ctrl and HD cortical neurons with and without Branaplam. **d** Agarose gel of PCR amplifying the HTT location of interest in fibroblasts with and without Branaplam. Lower band (88 bp) represents regular transcript without exon inclusion, upper band (203 bp) indicates integration of novel exon. **e** Bar plot illustrating quantification of agarose gel. Depicted is the ratio of densitometric quantification of included vs. excluded HTT transcript in fibroblasts of Ctrl (greens) and HD (purples) with (light shades) and without (dark shades) Branaplam treatment. Statistics: 2-way ANOVA (DMSO vs. Branaplam: $P < 0.0001$; Ctrl vs. HD: $P = 0.5954$; interaction: $P = 0.3139$). Bars: median $\pm$ IQR. **f** Agarose gel of PCR amplifying the HTT location of

interest in cortical neurons with and without Branaplam. Lower band (88 bp) represents regular transcript without exon inclusion, upper band (203 bp) indicates integration of novel exon. **g** Bar plot illustrating quantification of agarose gel. Depicted is the ratio of densitometric quantification of included vs. excluded HTT transcript in cortical neurons of Ctrl (greens) and HD (purples) with (light shades) and without (dark shades) Branaplam treatment. Statistics: two-way ANOVA (DMSO vs. Branaplam: $P = 0.0003$; Ctrl vs. HD: $P = 0.7070$; interaction: $P = 0.5633$). Bars: median $\pm$ IQR. **h, i** Bar plot illustrating HTT RPKM values in fibroblasts (**h**) and cortical neurons (**i**) of Ctrl (greens) and HD (purples) with (light shades) and without (dark shades) Branaplam treatment. Fibroblasts statistics: two-way ANOVA (DMSO vs. Branaplam: $P = 0.0005$; Ctrl vs. HD: $P = 0.0777$; interaction: $P = 0.2603$). Cortical neuron statistics: two-way ANOVA (DMSO vs. Branaplam: $P = 0.0101$; Ctrl vs. HD: $P = 0.6817$; interaction: $P = 0.2631$). Bars: median $\pm$ IQR. Source data are provided as a Source Data file.

cells detach. FGM was added to the detached cells, the cell suspension was transferred to a centrifugation tube and processed for 5 min at $300 \times g$ RT. The supernatant was removed, cells were resuspended in FGM and plated on a new polystyrene cell culture flask.

## iPSC generation and culture

For iPSC generation, skin biopsies of study participants were obtained. iPSCs were generated from fibroblasts using the CytoTune iPS 2.0 Sendai Reprogramming Kit (Thermo Fisher Scientific) according to the manufacturer's instructions. Therefore, cell lines were transduced with Sendai virus containing four reprogramming factors c-MYC, KLF4, OCT3/4, and SOX2.

After generation, iPSCs were cultured in human stem cell media StemMACS iPS-Brew XF (Miltenyi Biotec) supplemented with 100 U/mL penicillin/streptomycin on 4 mg/ml Geltrex (Gibco™) coated polystyrene cell culture plates. Medium was changed every other day. When cell cultures reached 70–80% confluency, cells were passaged. Afterward, iPSCs were washed once with DMEM/F12 (Gibco™) and incubated with Gentle Cell Dissociation Reagent (Stemcell technologies) for 5 min at room temperature (RT). Gentle Cell Dissociation Reagent was aspirated and StemMACS iPS-Brew XF supplemented with 100 U/mL penicillin/streptomycin was added. Corning® Cell Lifter was used to detach hiPSCs from the cell culture plate. iPSCs were transferred to a new Geltrex-coated plate.

## Cortical differentiation

iPSCs were differentiated into cortical neurons using a previously reported protocol[9]. In brief, iPSCs were maintained as described above. iPSCs were dissociated into a single-cell suspension upon 70–80% confluency. Cells were washed once with PBS w/o Mg2+/Ca2+ and were incubated with Accutase for 5 min at 37 °C. Cells were washed with DMEM/F12, centrifuged for 3 min at $300 \times g$ at RT, and resuspended in StemMACS iPS-Brew XF supplemented with 10 µM ROCK inhibitor. Cells were seeded on Geltrex-coated plates with the desired density of 300,000 cells per cm$^2$ and incubated for 24 h at 37 °C, 5% CO$_2$. After cells reached confluency the next day, the medium was changed to neural maintenance medium (NMM: DMEM/F-12, neurobasal/B-27/N2, 100 µM GlutaMAX, 100 µM non-essential amino acids, 50 µM 2-mercaptoethanol, 1× penicillin–streptomycin) supplemented with dual-SMAD inhibitors (NIM: 10 µM SB431542, 100 nM LDN193189) to promote neural induction. On day 12, cells differentiated into a neuroepithelial sheet and were further passaged. The cell sheet was gently washed with DMEM/F-12 and incubated for 5 min with Collagenase V (2 mg/ml) at 37 °C for 5 min. The cell sheet was gently washed twice with DMEM/F-12 and finally detached with a 5 ml serological pipette in NIM and gently resuspended into smaller pieces. Cells were passaged in a 1:2 ratio on Geltrex-coated plates. Medium was changed to NMM the next day. Upon appearance of neural rosettes, medium was changed for 2 days in NMM supplemented with 20 ng/ml

FGF2 to promote neural stem cell proliferation. On day 19, cells were further passaged and maintained in NMM with medium changes every second day. On day 30, cells were finallly single-cell passaged with Accutase with the desired density of 50,000 cells per cm$^2$. Cells were maintained in NMM for neuronal differentiation with medium changes twice a week till day 35 (Fig. 1) or day 50 (Figs. 2–6).

## Branaplam treatment

Branaplam was reconstituted in DMSO with a concentration of 5 M. Branaplam was supplemented to the cell culture media (FGM, StemMACS iPS-Brew XF or NMM) with a final concentration of 0.46–1000 nM and 0.002% DMSO. Supplemented medium was changed every 24 h for a total of 72 h.

## Immunofluorescent staining

Cells were fixed in 4% paraformaldehyde (PFA) for 20 min at RT and subsequently washed 3× with PBS each. The cells were permeabilized using 0.1% Triton-X-100 and in PBS for 20 min at RT. Then, cells were blocked in 0.3% Triton-X-100 and 3% donkey serum in PBS for 1 h at RT. Afterward, cells were incubated with primary antibodies (rat anti-CTIP2: ab18465, Abcam, 1:500; mouse anti beta-III-Tubulin: G7121, Promega, 1:1000; rabbit anti-PAX6, 901301, BioLegend, 1:200; mouse anti-Nestin, MAB5326, Millipore, 1:500) at 4 °C overnight. After washing, incubation with secondary antibodies and nuclei staining using 1 µg/ml DAPI was performed. The slides were mounted using ProLong(r) Antifade (Invitrogen) solution. Imaging was performed with a Zeiss Laser scanning 780 inverted confocal microscope.

## FACS analysis

For flow cytometry, cells were dissociated using Accutase for 30 min at 37 °C and resuspended in FC buffer (2% FCS, 0.01% sodium azide in PBS). Cells were dispensed into 15-ml tubes (Sarstedt) at 500,000 cells per tube. For intracellular antigens, cells were fixed and permeabilized using 100 µl BD Fixation/Permeabilization Solution (BD Bioscience) for 10 min, then 1 ml of BD Perm/Wash Buffer was added, cells were incubated for 5 min and subsequently centrifuged at $300 \times g$ for 3 min. For intracellular staining of cortical progenitors anti-PAX6-APC (130-123-267, Miltenyi Biotech, 1:100) and anti-NESTIN-PerCp-Cy5.5 (561231, BD Bioscience, 1:100) for an additional 30 min. After a wash step, cells were resuspended in 350 µl FACS buffer. For intracellular staining of neurons, cells were stained using anti-bIII-Tubulin-AF405 (NB600-1018AF405, NovusBio, 1:100) or anti-CTIP2-FITC (ab123449, Abcam, 1:100) for 30 min. Additional controls included applying an antibody solution without one antibody in the full cocktail ("minus 1 control") and were used to determine potential bleed through of the fluorophores. The flow cytometry experiments were performed with a Cytoflex S machine (laser 405 nm, 488 nm, 561 nm and 638 nm; Beckman Coulter) and analyzed with the CytExpert 2.4 software.

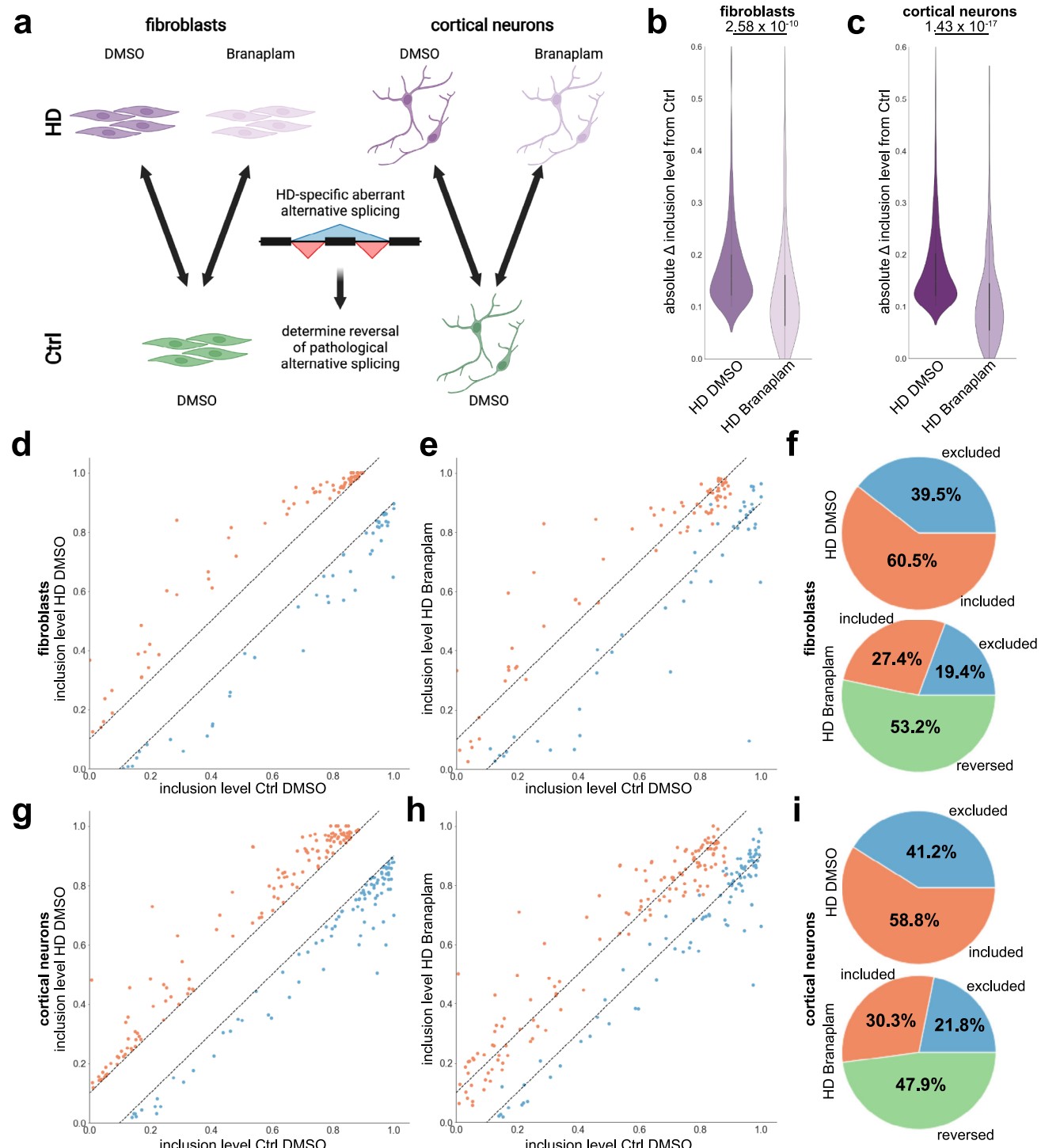

To determine cell death via FACS, we used a commercially available kit that uses a fluorescent 660-DEVD-FMK caspase-3/7 inhibitor reagent (ab270785, Abcam) and a fixable cell permeability dye (Live-or-Dye, 32008-T, Biotium). The caspase assay and Live-or-Dye assay reagents were dissolved in 50 μl DMSO, respectively and aliquoted and stored at −20 °C. For the assay, cortical neurons were grown in 24-well plates. At the day of analysis, media was aspirated from the plate and 150 μl DMEM/F12 + Glutamax containing 0.48 μl 660-DEVD-FMK caspase-3/7 inhibitor reagent and 0.15 μl Live-or-Dye assay were applied. After incubation for 45 min at 37 °C. Cells were dissociated, fixed, and stained as stated above. To precisely assess bleed through, single incubation controls (either with 660-DEVD-FMK caspase-3/7 inhibitor reagent or Live-or-Dye assay) were used. The number of Casp3/7+Live-

or-Dye- cells vs. Casp3/7-Live-or-Dye- were determined in CTIP2+ and beta-III-Tubulin+ cells.

In total, 500,000 NPCs were seeded on GELTREX-coated plates on day 26 of differentiation and treated with Branaplam or DMSO as described before. For EdU incorporation analysis, NPCs were treated with 30 μM EdU for 120 min at 37 °C in 5% CO$_2$. Cells were dissociated using Accutase for 30 min at 37 °C and resuspended in FC buffer (2% FCS, 0.01% sodium azide in PBS). Cells were dispensed into 5 ml tubes (Sarstedt) at 500,000 cells per well. For intracellular antigens, cells were fixed and permeabilized using 100 μl BD Fixation/Permeabilization Solution (BD Bioscience) for 10 min, then 1 ml of BD Perm/Wash Buffer was added, cells were incubated for 5 min and subsequently centrifuged at 300×g for 3 min. EdU-incorporated cells were stained

**Fig. 6 | Branaplam improves alternative splicing pathology in HD fibroblasts and cortical neurons. a** Paradigm illustrating samples used to determine improvements in AS pathology. Created with BioRender.com. **b, c** Violin plot showing absolute inclusion level differences (Ctrl DMSO − HD DMSO (dark shade) and Ctrl DMSO − HD Branaplam (light shade)) of HD AS events detected in all fibroblast (**b**) or cortical neuron (**c**) samples. Significance is calculated with Wilcoxon rank-sum test using scipy.stats. **d** Scatter plot illustrating inclusion levels of HD alternative splicing events detected in all fibroblast samples in Ctrl DMSO (x) and HD DMSO (y). Dashed lines mark corridor of absolute inclusion level difference <0.1. Red: events included in HD fibroblasts, blue: events excluded in HD fibroblasts. **e** Scatter plot illustrating inclusion levels of HD alternative splicing events detected in all fibroblast samples in Ctrl DMSO (x) and HD Branaplam (y). Dashed lines mark corridor of absolute inclusion level difference <0.1. Red: events included in HD fibroblasts, blue: events excluded in HD fibroblasts. **f** Pie charts quantifying the percentage of the number of events changed in HD fibroblasts that are excluded (blue), included (red) or rescued (green) (within absolute inclusion level difference corridor <0.1) in HD DMSO (top chart) and HD Branaplam (bottom chart). **g** Scatter plot illustrating inclusion levels of HD alternative splicing events detected in all cortical neurons samples in Ctrl DMSO (x) and HD DMSO (y). Dashed lines mark corridor of absolute inclusion level difference <0.1. Red: events included in HD cortical neurons, blue: events excluded in HD cortical neurons. **h** Scatter plot illustrating inclusion levels of HD AS events detected in all cortical neurons samples in Ctrl DMSO (x) and HD Branaplam (y). Dashed lines mark corridor of absolute inclusion level difference <0.1. Red: events included in HD cortical neurons, blue: events excluded in HD cortical neurons. **i** Pie charts quantifying percentage of number of events changed in HD cortical neurons that are excluded (blue), included (red) or rescued (green) (within absolute inclusion level difference corridor <0.1) in HD DMSO (top chart) and HD Branaplam (bottom chart).

according to manufactures protocol (BaseClick, BCK-FC594-50), followed by staining of anti-SOX2-PerCp-Cy5.5 (BD Bioscience, 561506, 1:100) for 15 min at room temperature. After a wash step, cells were resuspended in 1000 µl FACS buffer buffer containing DAPI (1 µg/ml).

**Protein extraction**

Lysis buffer (150 mM NaCl, 20 mM Tris-HCl (pH 7.5), 1 mM EDTA, 1 mM EGTA, 1% Triton-X-100, 10 mM NaF, 1 mM PMSF, 1× Phosphatase Inhibitor, and 1× Protease Inhibitor in dH$_2$O) was added to the cells for 30 min at 4 °C. Cell lysate was transferred into a tube and centrifuged for 15 min at 2000×$g$ at 4 °C. Supernatant was transferred to a new centrifugation tube and stored at −80 °C. Protein concentration was determined using BCA assay kit (Thermo Fisher Scientific) measuring with CLARIOStar Plus (BMG Labtech).

**HTT assay**

Meso-Scale-Discovery (MSD) assays to measure total and mutant Huntingtin protein levels were performed by Evotec SE, Hamburg[10]. A first antibody (2B7) is used to capture HTT, and a second antibody (D7F7 or MW1) is used to detect and quantify HTT vial a SULFO-TAG. The 2B7 antibody binds to the first 17 amino acids of HTT. The D7F7 antibody binds downstream of the poly-Q tract, at a single epitope in the center of HTT. In combination, 2B7 and D7F7 can be therefore used to detect total HTT levels regardless of their poly-Q length (tHTT). The MW1 antibody is widely used to detect mutant HTT. Together with 2B7, they are currently used to detect mutant HTT in the CSF of HD patients[22,23] and therefore used in HTT lowering clinical trials. MW1 has a higher avidity to mHTT due to bivalent binding of the antibody at sites with elongated poly-Q. Thus, this poly-Q-binding antibody do not specifically, but preferentially recognize mHTT and it can be expected that the 2B7/MW1 assay will result in a higher signal for mutant, expanded HTT[23,24]. However, it is important to note that also non-mutant HTT with shorter poly-Q length is recognized with this antibody combination[23,24].

The MSD assay plate was coated with 5ug/ml of the N-terminally binding HTT antibody 2B7 (#CH03023, Coriell) in coating buffer (15 mM Na2CO3, 35 mM NaHCO3) overnight. The next day, the plate was washed 3× in wash buffer (0.2% (v/v) Tween 20 in DBPS), blocked for 1 h at RT shaking at 350 rpm (2% (w/v) Probumin in wash buffer) and subsequently washed 3× again. The MSD plate was then incubated with the protein sample derived from the various cells (10 µl sample/well) for 1 h at RT shaking at 350 rpm. In parallel, a standard of defined concentrations of recombinant human HTT with 23Q or 73Q was applied. After incubation, the plate was washed 3x in wash buffer. Next, 10 µl of the detection antibodies were added to the MSD plates: 0.5ug/ml D7F7 antibody (#5656, Cell Signaling) for tHTT detection, or 5ug/ml MW1 antibody (#MABN2427, Sigma-Aldrich) binding the polyQ region in exon 1 for mutant HTT detection. MW1 was used directly labeled with a SULFO-Tag and incubated for 1 h at RT shaking at 350 rpm and subsequently washed 3x. For D7F7, after three washes, a SULFO-Tag-labeled anti-rabbit secondary antibody (MSD) was incubated for 1 h at RT, and the plate was subsequently washed 3×. MSD read buffer was added to the plate. If the detection antibody binds to the sample in close proximity to the MSD plate an electrochemiluminescent signal is emitted and detected at 620 nm. The total and mutant HTT levels were calculated according to the generated standard curves and normalized to protein input. The signal values were back-calculated to the standard that was run in parallel (Q23 for 2B7/D7F7 assay and Q73 for 2B7/MW1 assay), resulting in MSD signal values of a sample equivalent to a certain HTT Q23 or HTT Q73 concentration. The assay does not allow to compare the numerical results from the total HTT assay and the mutant HTT assay with each other. The numerical values from both assays cannot be compared by mathematical addition as they are two separate assays with different antibodies that have not identical binding properties (e.g., D7F7 binds once, MW1 can bind multiple times depending on poly-Q length).

**Toxi light assay**

Cytotoxicity was measured during Branaplam treatment using the ToxiLight Bio assay kit (Lonza) according to the manufacturer's instructions. Therefore, the supernatant was collected after 72 h of Branaplam treatment. The positive control was a supernatant of untreated cells incubated with 10% Triton-X-100 for 20 min at 37 °C. Triplicates 20 µl/sample were transferred to a 96-well plate. In total, 100 µl of adenylate kinase detection reagent (ToxiLight Bio assay kit, Lonza) was added and incubated for 5 min at RT. The resulting luminescence was measured by the CLARIOStar Plus (BMG Labtech).

**Soluble/insoluble protein fractionation**

The cortical neuron samples were scraped off the plate in DBPS and transferred to a 1.5-ml Eppendorf tube and centrifuged and subsequently, dry ice flash frozen and stored at −80 °C until further processing. The cells were lyzed in 150 µl ice-cold RIPA buffer. The cell lysate was sonicated for five minutes with 30 s on/off on a low-intensity level using a Bioruptor. Afterward, the lysate was centrifuged at 100,000×$g$ for 30 min at 4 °C. This supernatant represents the soluble protein fraction. The pellet was washed with ice-cold RIPA once and then resuspended in 75 µl urea buffer (7 M urea, 3 M thiourea, 4% CHAPS, 30 mM Tris). Subsequently, the suspension was sonicated using a Bioruptor under the same conditions as described before. The sonicated suspension is centrifuged at 100,000×$g$ for 30 min at 4 °C again. The resulting supernatant reflects the insoluble fraction. The protein content in each fraction was quantified with bicinchoninic acid (BCA) assay. Equal concentrations were applied, and loading buffer and DTT (final concentration of 100 mM) was added. The samples were incubated at 55 °C for 30 min. The samples were then used for western blotting.

## Western blot

All immunoblots were run on 4–12% Bis-Tris gels with NuPAGE MOPS running buffer for 90 min at 180 V. Proteins were transferred to a PVDF membrane with 10% methanol in NuPAGE transfer buffer at 30 V overnight at 4 °C. The membrane was then blocked for 1 h in 5% dry milk in TBS-T and primary antibody (RBFOX2: A300-864A, Bethyl Laboratories Inc., 1:1000; ILF3: A303-651A-T, Bethyl Laboratories Inc., 1:2000; QKI: A300-183A-T, Bethyl Laboratories Inc., 1:2000; U2AF2: A303-667A-T, Bethyl Laboratories Inc., 1:2000; TIAL1: RN059PW, MBL international, 1:1000) was incubated overnight at 4 °C. Afterwards, the membrane was washed twice for approximately seven minutes in TBS-T and then incubated for one hour with the secondary HRP-conjugated antibody for one hour at room temperature. The membrane was washed three times with TBS-T and incubated in the dark with ECL solution. Film development was performed in the dark with various exposure times. For the quantification of western blots, densitometric analysis was performed using Fiji. The signal was normalized to the corresponding signal in the Coomassie-stained gels reflecting the total protein amount.

## RNA extraction and HTT novel exon PCR

RNA was extracted using the RNeasy kit (Qiagen) according to the manufacturer's instructions. RNA concentrations were measured using a NanoDrop. The GoScript Reverse Transcriptase cDNA Synthesis kit (Promega) was used to generate cDNA from fibroblasts and cortical neurons using random primers. RNA was mixed with random primers and incubated for 5 min at 70 °C and placed on ice for 5 min. The remaining reaction mix was added and incubated for 5 min at 25 °C, followed by 1 h 42 °C extension period and a 15 min 70 °C inactivation. The GoTaq 2× Mastermix (Promega) was used to amplify novel exon inclusion in HTT, amplifying 0.5 µl of the template with 1 µl of fwd primer (100 µM stock, GTCATTTGCACCTTCCTCCT) and 1 µl rev primer (100 µM stock, TGGATCAAATGCCAGGACAG), 5 µl Mastermix and 2.5 µl DNase/RNase-free water. Primer sequences were obtained from the Novartis patent (WO2021084495A1). The mix was amplified with the following conditions: 95 °C for 3 min, and 34 cycles of 95 °C for 30 s, 60 °C for 20 s, and 72 °C for 60 s. A final extension of 72 °C for 5 min was added at the end. The products were run on a 2% Agarose gel with RotiGel stain. (Carl Roth GmbH) at 125 V. A random selection of 88 bp and ~200 bp bands in Ctrl and HD was cut out and purified to verify their correct identity by Sanger sequencing.

## RNA sequencing

A total of 500 ng per sample were sent for RNA sequencing to Azenta Life Sciences (Genewiz Leipzig, Germany) for 150 bp paired-end sequencing with Poly-A selection. For fibroblasts, four Ctrl samples and four HD samples with DMSO or Branaplam treatment were sent and sequenced at a depth of >20 million reads in each sample. For iPSC-derived cortical neurons, three Ctrl samples and three HD samples with DMSO or Branaplam treatment were sent and sequenced at a depth of >37 million reads in each sample. After obtaining the fastq files, adapters were trimmed using Trimmomatic[25] and aligned to the human genome (GRCh38) using STAR[26]. In every sample, >90% of reads mapped uniquely to the human genome. Reads were assigned to genes in the gencode annotation (version 26) using the featureCounts module within the Subread package[27]. Reads Per Kilobase of transcript, per Million mapped reads (RPKM) were calculated from the obtained counts to normalize for gene expression.

## Alternative splicing analysis

For differential splicing rMATS (version 4.2.0)[28] was used with the novelSS flag to identify non-annotated exons. The gencode annotation (version 26) was used to define known exons. The output files considering only the junction counts were used for further analysis. A negative value of the InclusionLevelDifference reflects an inclusion of a given exon in the samples of the target condition and a positive value

of the InclusionLevelDifference reflects an exclusion of a given exon in the samples of the target condition. Subsequently, the files from the different splice types (cassette exon, A5SS, A3SS, RI and MXE) were combined into one file.

All downstream analyses were performed in Python 3. Only exon junctions that were covered with at least ten counts in each sample of a given dataset were considered. A unique index was generated, referring to a specific AS event with the aim to identify the identical exon junction in separate rMATS analyses. An exon was called as differentially alternatively spliced in each dataset if the FDR was below 0.05 and the absolute value of the InclusionLevelDifference was more than 0.1. The overlap of differentially spliced events was visualized with the Venn function in matplotlib library.

For k-means clustering, the k-means method from the sklearn.cluster module from sciki-lean was used (specifications: init = "ranodm", n_clusters = 10, n_init = 10, max_iter = 300, random_state = 42). AS events significant in any of the four comparisons (fibroblasts-Ctrl DMSO vs Branaplam, fibroblasts-HD DMSO vs Branaplam, cortical neurons-Ctrl DMSO vs Branaplam, cortical neurons-HD DMSO vs Branaplam) datasets were clustered into 10 clusters according to the inclusion value differences in the respective dataset. Exon junctions that were not detected with a sufficient number of reads were masked and visualized in black. The inclusion level difference of each cluster was additionally visualized with violin plots.

In order to determine the overlap of Branaplam-induced events of this study to Branaplam-induced events reported by Monteys and colleagues in HEK293 cells[15], pybedtool was used to compare overlap of genomic locations.

To determine the effect of aberrant AS reversal upon Branaplam treatment, the individual inclusion values were used from HD-DMSO, HD-Branaplam, and Ctrl-DMSO samples in fibroblasts and cortical neurons, respectively. Only significant HD alternative splicing events (Ctrl DMSO vs. HD DMSO) in a respective cell type that were detected in all samples analyzed in a cell type (>10 reads in every single sample) were used. The reversal of aberrant AS was investigated in a quantitative and qualitative manner. For quantitative measurement, the absolute inclusion value difference was calculated by subtracting HD-DMSO or HD-Branaplam inclusion values from Ctrl-DMSO inclusion values and taking the absolute value. The statistical significance of the absolute inclusion level difference was determined using scipy.stats.ranksums. For qualitative measurement, the mean inclusion values of Ctrl-DMSO and HD-DMSO and HD-Branaplam samples in each cell type were also visualized in a scatter plot. A reversal of aberrant AS was determined if the inclusion level differences in HD-Branaplam samples dropped below an absolute value of 0.1.

## RBP enrichment

In order to determine RNA-binding proteins that are enriched in alternatively spliced events in HD, we made use of the ENCODE database and their eCLIP-seq datasets. We downloaded eCLIP seq peak files aligned to GRCh38 with the Irreproducible Discovery Rate (IDR) peaks (released by November 2021). A peak was considered significant if negative $\log_{10}(P \text{ value}) \geq 3$ and the $\log_2(\text{fold change}) \geq 3$. To determine if an eCLIP-seq peak was present in an exon junction in HD, the rMATS output (fibroblasts Ctrl-DMSO vs HD-DMSO or cortical neurons Ctrl-DMSO vs HD-DMSO) of interest was converted into a bed format encompassing the region starting from the upstream exon start to the downstream exon end. The rMATS bed was intersected with the significant eCLIP-seq peak file using pybedtools (-u True). The statistical significance of the enrichment was computed using hypergeometric test with all events that passed the coverage threshold as the background. In order to determine the significance of RBPs that are rather enriched in rescued HD AS events vs not rescued HD AS events by Branaplam, Fisher's exact test was applied to compare RBP binding in the two sets of AS events.

## Kmer enrichment

To determine the sequence preferences of Branaplam-induced AS sites at the 3′ and 5′ splice site, we calculated kmer enrichments (4mer, 6mer, and 8mer) 5b upstream and downstream of the 3′ and 5′ splice site, respectively. As a background, we also calculated the 5b upstream and downstream of the 3′ splice site of the respective upstream exon and the 5′ splice site of the respective downstream exon. Kmers were counted with Kvector (https://github.com/olgabot/kvector), and significance was determined with Fisher's exact test using scipy.stats.

## Statistical analysis

GraphPad Prism 9 was used to visualize data and calculate statistics for pair-wise and grouped analyses (HTT protein measurements, toxilight assay, FACS quantification, densitometric quantification of HTT PCR, HTT RPKM values). DMSO samples and their respective Branaplam samples were considered as paired. Normal distribution was assessed with Shapiro–Wilk test. When comparing two conditions, Welch's test was used if normal distribution was confirmed and Mann–Whitney test was used for non-normally distributed data. When comparing multiple groups (e.g., different Branaplam concentrations), one-way ANOVA with Geisser–Greenhouse correction was used if normal distribution was confirmed and Friedman test was used for non-normally distributed data with Dunnett's or Dunn's post hoc test, respectively, to identify differences between individual groups. For grouped analyses (e.g., DMSO vs. Branaplam in Ctrl vs. HD), two-way ANOVA was used. The statistical test used for calculating the significance of each graph is indicated in the figure legend. A $P$ value ≤ 0.05 was considered as significant.

## Reporting summary

Further information on research design is available in the Nature Research Reporting Summary linked to this article.

## Data availability

Due to the European General Data Protection Regulation and specifically patient consent of study participants who donated biological material used for the generation of the RNA-seq datasets, access will be granted via the European Genome Phenome Archive (EGA). The fibroblast RNA-seq dataset is available under the accession number EGAD00001008807 and the cortical neuron RNA-seq dataset is available under the accession number EGAD00001008808. Additional data, information, and materials that support the findings of this study are available from the corresponding author upon reasonable request. Source data are provided with this paper.

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

## Acknowledgements

We would like to thank all patients and control individuals who donated biological material to conduct this study. We thank Naime Denguir, M.Sc., Dr. Pooja Gupta and Dr. Wolfgang Krebs (CUBIDA core unit) for technical support. Data analysis was performed at the servers hosted by

the "Data Integration Center" (DIC) and supported by the "Core Unit for Bioinformatics, Data Integration and Analysis" (CUBiDA), Universitätsklinikum Erlangen, Erlangen, Germany. J.S. is a graduate student of the research training group 2162 "Neurodevelopment and Vulnerability of the Central Nervous System" of the Deutsche Forschungsgemeinschaft (DFG 270949263/GRK2162 (B.W. and J.W.)). This study was further supported by the Interdisciplinary Center for Clinical Research (IZKF) at the University Hospital Erlangen (Junior Project [J88], F.K.), the Bavarian Ministry of Science and the Arts in the framework of the ForInter network (B.W. and J.W.), the TreatHSP consortium (BMBF 01GM1905B, B.W. and J.W.), the German Research Foundation, DFG WI 3567/2-1 (B.W.), Universitaetsstiftung Medizin (JW) and the IZKF advanced project E30 (B.W. and J.W.).

## Author contributions

F.K. J.S., T.B., J.W., and B.W. designed experiments. F.K. performed all computational and statistical analyses. J.S., S.P., and M.F. reprogrammed fibroblasts into iPSC. F.K., J.S., T.B., J.S., T.B., and S.R. performed cell culture, differentiations, IF stainings, RBP solubility analyses, FACS, and western blot. A.W. and I.L. performed HTT MSD assays. H.M. performed HTT splicing RT-PCR. U.H. conducted CAG repeat analysis. Z.K. and J.W. recruited patients and controls. F.K., J.W., and B.W. acquired funding. J.W. and B.W. supervised the work. F.K. made figures and wrote the original draft of the manuscript. F.K., J.S., T.B., J.W., and B.W. edited the manuscript. All authors read and agreed to the manuscript.

## Funding

## Competing interests

The authors declare no competing interests.

## Additional information

Florian Krach [ID][1], Judith Stemick [ID][2], Tom Boerstler [ID][1], Alexander Weiss[3], Ioannis Lingos[3], Stephanie Reischl [ID][1], Holger Meixner[2], Sonja Ploetz[2], Michaela Farrell[1], Ute Hehr[4], Zacharias Kohl[5], Beate Winner [ID][1,6,7] ✉ & Juergen Winkler [ID][2,6,7] ✉

[1]Department of Stem Cell Biology, University Hospital Erlangen, Friedrich-Alexander University of Erlangen-Nürnberg (FAU), Erlangen, Germany. [2]Department of Molecular Neurology, University Hospital Erlangen, Friedrich-Alexander University of Erlangen-Nürnberg (FAU), Erlangen, Germany. [3]Evotec SE, Hamburg, Germany. [4]Zentrum für Humangenetik Regensburg, Regensburg, Germany. [5]Department of Neurology, University of Regensburg, Regensburg, Germany. [6]Center for Rare Diseases Erlangen (ZSEER), University Hospital Erlangen, Friedrich-Alexander University of Erlangen-Nürnberg (FAU), Erlangen, Germany. [7]These authors contributed equally: Beate Winner, Juergen Winkler. ✉e-mail: beate.winner@fau.de; juergen.winkler@uk-erlangen.de

