## [Peer Review File · Nature Communications]

An alternative splicing modulator decreases mutant HTT and improves the molecular fingerprint in Huntington's disease patient neuronsREVIEWER COMMENTS

Reviewer #1 (Remarks to the Author):

Review for "An alternative splicing modulator decreases mutant HTT and improves the molecular fingerprint in Huntington's disease patient neurons"

Overall observations:

The manuscript, as a whole, is quite interesting and represents a well-researched series of studies to understand the role of alternative splicing in both the disease pathology of HD and the mechanism-of-action (MoA) of Branaplam. The science and data are performed well – and with appropriate controls, etc. – and represent important observations. There are one major and several minor points with the manuscript that I would recommend be addressed before publication.

Major point:

The authors provide interesting data to support the importance of alternative mRNA splicing in both the MoA of Branaplam and the pathology of HD. The descriptions of both datasets focus on specific details of the observed AS changes (e.g., identification of the AGAGTAAG sequence at the 5' splice site of the unannotated HTT exon). The intersection between these two related, and interconnected, phenotypes is left with a very minimal exploration in both the Results and Discussion sections. I believe this is one of the most important findings in the current manuscript – and could benefit from a more fulsome exploration (e.g., how does Branaplam-mediated decrease in mHTT lead to activity of U2AF2 – a splicing factor that is impacted by mHTT expression). The data are already gathered, and this would just require some additional description in the text (e.g., Results to include a comparison between the AS factor sites restored – is there an enrichment in some, but not all? – or are there genes with complimentary / competing AS events driven by mHTT and Branaplam?).

Minor points:

The English grammar and syntax in the manuscript are stilted and could benefit from some "style editing" to improve comprehension and readability.

For example – the two sentences between lines 30-35 in the abstract could be better constructed for clarity. Specifically, the second sentence (beginning with Amongst...) seems to be more structured as a dependent clause of the first sentence.

There are other examples throughout the manuscript (e.g., the sentence from lines 59-61. This should be rewritten to include something like "[restore] mHTT induced [aberrant splicing] of other transcripts" for clarity; where bolded text is added to the existing text shown in brackets.)

Summary:

As noted above, the current version of the manuscript could benefit from both a deeper exploration and description of the intersection of the AS events driven by mHTT and Branaplam as well as some style / copyediting. With these changes, I would recommend the manuscript for publication.

Reviewer #2 (Remarks to the Author):

Huntington's disease (HD) is a dominant, monogenic, progressive neurodegenerative disorder caused by poly-glutamine (poly-Q) expansion in the HTT protein. Branaplam (NVS-SM1, LMI070) is an orally dosed small-molecule drug that previously underwent a clinical trial for treatment of spinal muscular atrophy (SMA), although it was withdrawn after a clinical hold, and Novartis is currently exploring it instead for treatment of HD. In this manuscript, the authors report that Branaplam treatment promotes downregulation of mutant and total HTT mRNA and protein.

The authors first show that they can measure mutant HTT (mHTT) with poly-glutamine expansion in

patient-derived fibroblasts, iPS cells, and iPS-derived neurons, using an antibody specific for the long poly-Q expansion (Fig. 1). Then, they report differential splicing events between HD and healthy controls, associated with reported UV crosslinking sites for several RNA-binding proteins that may be the cause of aberrant splicing (Figure 2).

Next, the authors treated control and HD patient-derived cells with Branaplam. The drug significantly lowered the abundance of total HTT (tHTT) and mHTT in fibroblasts and iPS cells, without inducing apoptotic caspase 3/7 activation.

RNA-seq analysis of Branaplam-treated fibroblasts and cortical neurons revealed inclusion of non-annotated exons and sequence motifs at the 3' and 5' splice sites associated with these events. Among the 55 genes with Branaplam-induced novel exons, HTT showed inclusion of a frameshifting exon, leading to a decrease in mRNA levels due to NMD (Figure 4). The novel HTT exon is 115-nt long, has two stop codons, and was validated by RT-PCR in all cohorts and cell types tested (Figure 5).

The authors then tested if Branaplam treatment reverses the aberrant splicing events in HD patient-derived cells. Treatment reversed about half of the splicing events characteristic of HD (Figure 6).

The manuscript clearly shows that Branaplam can partially reverse the molecular phenotype of HD patient-derived differentiated neurons. However, the concept of treating HD with Branaplam largely overlaps with Keller et al. (ref. 15) published earlier this year in Nature Communications. Splicing alterations in HD were also previously reported in post-mortem cortex and striatum (refs. 11 and 12), although the mechanism and significance remain unknown. It was not shown in the present manuscript whether the aberrant splicing of six key pathogenic genes (CCDC88C, KCTD17, SYNJ1, VPS13C, TRPM7 and SLC9A5) listed in ref. 12 were reversed by treatment. Detailed studies of the RBPs listed in the manuscript (RBFOX2, TIA1, U2AF2) and mHTT itself, such as expression-level alterations, subcellular localization, and PTMs, and reversal of individual splicing events by knockout or overexpression of each of these RBPs would improve the manuscript by providing some mechanistic insight about the link between the expanded poly-Q stretch and aberrant splicing.

The manuscript only cites Keller et al. as targeting SH-SY5Y cells, although it also describes the effect of Branaplam in vivo in a mouse model of HD. These recently published in vivo experiments support the potential of Branaplam as a treatment for HD.

The off-target splicing effects of Branaplam were previously reported by Monteys et al (2021) Nature 596: 291-295. This reference should be cited, and whether the splicing events detected overlap with those in the present study should be addressed.

The drug reverses some aberrant splicing events associated with HD, but it induces other aberrant splicing events. Although caspase 3/7 cleavage was not induced, this is a limited way to address potential toxicity.

Reviewer #3 (Remarks to the Author):

Several approaches to lowering the levels of the huntingtin protein (HTT) are currently in clinical trials for evaluation as disease modifying treatments for Huntington's disease (HD). This includes the small molecule splicing modulator, Branaplam, which results in the inclusion of exon 49a, leading to a frame shift, and nonsense-mediated decay of the HTT transcript. In this study, the authors have investigated the effects of Branaplam in HD patient derived cells including fibroblasts and iPSC derived cortical

neurons. First, they show that fibroblasts and cortical neurons recapitulate aberrant alternative splicing as a molecular fingerprint of HD. They demonstrate that Branaplam lowers total HTT and mutant HTT levels in fibroblasts, iPSCs, cortical progenitors, and neurons in a dose dependent manner at an IC50 consistently below 10nm without inducing cellular toxicity. Branaplam promotes inclusion of non-annotated novel exons, and they used an unbiased approach to identify the effects of Branaplam on splicing at the transcriptome level and the effects of this on gene expression. Branaplam was found to ameliorates aberrant alternative splicing in HD patients' fibroblasts and cortical neurons.

This is a timely study, and the effects of Branaplam on splicing throughout the transcriptome will be of great interest to those in the HD community and others interested in the actions of these splicing-modulator type drugs. Unfortunately, this paper is let down by the protein data, which cannot be presented in their current form.

Introduction

Line 41: Individuals with CAG repeats of 40 and above will develop HD, not 39 and above. The authors are correct in the following sentence when they state that alleles of 36-39 exhibit incomplete penetrance.

Protein data

Figure 1E-I

These data have been interpreted incorrectly. The 2B7-MW1 assay, to all intents and purposes, is specific to mutant HTT. Therefore, the 2B7-MW1 signal in the various HD and control cell lines indicated that mutant HTT is present in the HD lines and not in the controls. It does not indicate that the level of mutant HTT was increased in the HD lines as compared to the controls. Given that the total HTT assay measures both mutant and wild type, the fact that this is constant between the HD and control lines indicates that mutant HTT and wild type HTT levels are comparable.

The MSD data are very poor given that these are supposed to be clinical-grade GLP-approved assays.

Figure 3

The data in this figure do not make sense

e.g. panel D, untreated cells:

The total HTT assay (control and mutant) gives a concentration of 400 pmol/g for one HD cell line. Presumably, given that Figure 1I indicated that there was no difference in total HTT levels between control and HD cells, it can be assumed that 200 pmol/g of this is comprised of wild type HTT and 200 pmol/g of mutant HTT. Yet the mutant HTT assay indicated that there are only 25 pmol/g of mutant HTT in this line.

Similarly panel E indicates that the HD and control lines each have 100 pmol/g of total HTT (HD and wild type) and yet the mutant HTT assay indicates that the level of mutant HTT alone is 100 pmol/g.

Clearly the method of protein quantification that has been used in this paper gives confusing and contradictory data. It cannot be represented in this way as it is currently meaningless.

Point-by-point response to the reviewers

Reviewer comments highlighted in gray. Our answers to the reviewers are indented.

Reviewer #1 (Remarks to the Author):

Review for “An alternative splicing modulator decreases mutant HTT and improves the molecular fingerprint in Huntington’s disease patient neurons”

Overall observations:

The manuscript, as a whole, is quite interesting and represents a well-researched series of studies to understand the role of alternative splicing in both the disease pathology of HD and the mechanism-of-action (MoA) of Branaplam. The science and data are performed well – and with appropriate controls, etc. – and represent important observations. There are one major and several minor points with the manuscript that I would recommend be addressed before publication.

We thank the reviewer for this very positive general response.

Major point:

The authors provide interesting data to support the importance of alternative mRNA splicing in both the MoA of Branaplam and the pathology of HD. The descriptions of both datasets focus on specific details of the observed AS changes (e.g., identification of the AGAGTAAG sequence at the 5' splice site of the unannotated HTT exon). The intersection between these two related, and interconnected, phenotypes is left with a very minimal exploration in both the Results and Discussion sections. I believe this is one of the most important findings in the current manuscript – and could benefit from a more fulsome exploration (e.g., how does Branaplam-mediated decrease in mHTT lead to activity of U2AF2 – a splicing factor that is impacted by mHTT expression). The data are already gathered, and this would just require some additional description in the text (e.g., Results to include a comparison between the AS factor sites restored – is there an enrichment in some, but not all? – or are there genes with complimentary / competing AS events driven by mHTT and Branaplam?).

The reviewer addresses very interesting topics. We have performed the analyses as suggested. There almost no overlap between the Branaplam events and HD events. In fibroblasts no overlap was observed and only 4 Branaplam events were changed in HD

(2 in the same, 2 in the opposite direction). This suggests that Branaplam's effects on HD splicing are not direct but rather indirect e.g. through reduction of mHTT RNA and protein levels.

As suggested by the reviewer we also analyzed if rescued HD splicing is associated with a specific subset of the identified RBPs (Figure 2). We have performed this analysis for fibroblast and cortical neuron HD splicing by plotting the enrichment in the rescued vs. the non-rescued events (from Figure 6) and calculating statistical significance by Fisher's exact test. Most interestingly, the RBP QKI was rather associated with branaplam-rescued AS events in both ENCODE cell types (K562 and HepG2). We have added this finding to Extended Data Figure 6.

We have summarized the above mentioned findings in the results section as followed:

“Branaplam itself did not directly target the HD-specific AS events (Extended Data Figure 6A and B). The rescued AS events were preferentially bound by certain RBPs (Extended Data Figure 6C and D), e.g. rescued events cortical neurons were more prominently bound by QKI than non-rescued events (Extended Data Figure 6D). This suggests that Branaplam may revert HD AS events indirectly by reduction of mHTT RNA and protein levels consequently altering RBPs' functions.”

Minor points:

The English grammar and syntax in the manuscript are stilted and could benefit from some “style editing” to improve comprehension and readability. For example – the two sentences between lines 30-35 in the abstract could be better constructed for clarity. Specifically, the second sentence (beginning with Amongst...) seems to be more structured as a dependent clause of the first sentence. There are other examples throughout the manuscript (e.g., the sentence from lines 59-61. This should be rewritten to include something like “[restore] mHTT induced [aberrant splicing] of other transcripts” for clarity; where bolded text is added to the existing text shown in brackets.)

We have carefully revised the manuscript with a special focus on the mentioned sentences. We adapted grammar and syntax as suggested by the reviewer and changed the relevant sentences accordingly. We hope that this improved comprehension and readability.

Summary:

As noted above, the current version of the manuscript could benefit from both a deeper exploration and description of the intersection of the AS events driven by mHTT and Branaplam as well as some style / copyediting. With these changes, I would recommend the manuscript for publication.

Reviewer #2 (Remarks to the Author):

Huntington's disease (HD) is a dominant, monogenic, progressive neurodegenerative disorder caused by poly-glutamine (poly-Q) expansion in the HTT protein. Branaplam (NVS-SM1, LMI070) is an orally dosed small-molecule drug that previously underwent a clinical trial for treatment of spinal muscular atrophy (SMA), although it was withdrawn after a clinical hold, and Novartis is currently exploring it instead for treatment of HD. In this manuscript, the authors report that Branaplam treatment promotes downregulation of mutant and total HTT mRNA and protein.

The authors first show that they can measure mutant HTT (mHTT) with poly-glutamine expansion in patient-derived fibroblasts, iPS cells, and iPS-derived neurons, using an antibody specific for the long poly-Q expansion (Fig. 1). Then, they report differential splicing events between HD and healthy controls, associated with reported UV crosslinking sites for several RNA-binding proteins that may be the cause of aberrant splicing (Figure 2).

Next, the authors treated control and HD patient-derived cells with Branaplam. The drug significantly lowered the abundance of total HTT (tHTT) and mHTT in fibroblasts and iPS cells, without inducing apoptotic caspase 3/7 activation.

RNA-seq analysis of Branaplam-treated fibroblasts and cortical neurons revealed inclusion of non-annotated exons and sequence motifs at the 3' and 5' splice sites associated with these events. Among the 55 genes with Branaplam-induced novel exons, HTT showed inclusion of a frameshifting exon, leading to a decrease in mRNA levels due to NMD (Figure 4). The novel HTT exon is 115-nt long, has two stop codons, and was validated by RT-PCR in all cohorts and cell types tested (Figure 5).

The authors then tested if Branaplam treatment reverses the aberrant splicing events in HD patient-derived cells. Treatment reversed about half of the splicing events characteristic of HD (Figure 6).

The manuscript clearly shows that Branaplam can partially reverse the molecular phenotype of HD patient-derived differentiated neurons. However, the concept of treating HD with Branaplam largely overlaps with Keller et al. (ref. 15) published earlier this year in Nature Communications. Splicing alterations in HD were also previously reported in post-mortem cortex and striatum (refs. 11 and 12), although the mechanism and significance remain unknown. It was not shown in the

present manuscript whether the aberrant splicing of six key pathogenic genes (CCDC88C, KCTD17, SYNJ1, VPS13C, TRPM7 and SLC9A5) listed in ref. 12 were reversed by treatment. Detailed studies of the RBPs listed in the manuscript (RBFOX2, TIA1, U2AF2) and mHTT itself, such as expression-level alterations, subcellular localization, and PTMs, and reversal of individual splicing events by knockout or overexpression of each of these RBPs would improve the manuscript by providing some mechanistic insight about the link between the expanded poly-Q stretch and aberrant splicing.

We thank the reviewer for the careful review and interest in our manuscript, especially in alternative splicing as a mechanism of pathogenesis in HD. We share the enthusiasm about this interesting phenotype. As we previously referenced in the manuscript and the reviewer pointed out, alternative splicing was observed in postmortem tissue of HD patients. The added value of our data is that we are already able to identify this phenotype during lifetime of the patient in iPSC-derived neurons that recapitulate a very early disease stage. The novel information in our manuscript is that aberrant alternative splicing appears to be an early disease feature before signs of neurodegeneration are present. We have added a paragraph in the discussion stating the importance of this phenotype at early disease stages. Moreover, we particularly enlarged the comparison to other neurodegenerative disorder, especially ALS, where we and others were able to show that alternative splicing is not only observed at the end stage of disease but already present at early stages:

“Our iPSC-based model recapitulates aberrant alternative splicing as a feature of HD that has been previously observed in postmortem tissue ^{11,12}. Similarly, aberrant alternative splicing in iPSC cortical neurons is associated with binding of specific RNA-binding proteins ¹². Aberrant alternative splicing is an interesting phenotype in the context of neurodegenerative diseases and has been most frequently studied in ALS patients. Widespread AS changes are observed in postmortem tissue ^{16,17}. This phenotype was also observed in ALS iPSC models ^{14,18} that are driven by biochemical and cellular alterations in specific RNA-binding proteins ¹⁴.”

We have also thoroughly reanalyzed the alternative splicing and reversal by branaplam in the 6 suggested genes (CCDC88C, KCTD17, SYNJ1, VPS13C, TRPM7 and SLC9A5) described by Elorza et al. However, in our iPSC-derived model we are not able to detect

changes and reversal in those 6 events in HD (from rMATS outputs available in Source Data). One possible reason for this discrepancy may be our use of cortical neurons vs. the use of striatum by Elorza et al.. Another reason may be that at these very early stages of disease these key pathogenic events may not be present yet due to lack in aberrations in RBP solubility, localization, and function happening further downstream in the disease pathogenesis.

Regarding the functional description of the RBPs identified to be associated with aberrant alternative splicing in HD, we agree that additional profiling to elucidate cellular and biochemical origins for their disease-associated behavior in alternative splicing is interesting. We analyzed the gene expression of all significantly enriched RBPs in fibroblasts and cortical neurons (Extended Data Figure 2). We detected reduced U2AF2 and U2AF1 levels. Reduction in U2AF2 levels were also reported before by Elorza et al..

We have recently published a study investigated the association of RBP solubility and alternative splicing in iPSC-models of ALS and identified that the elevated insolubilities of newly identified RBPs is associated with aberrant alternative splicing in ALS (Krach et al., 2022). This lead us to investigate the RBP insolubility of the top 5 hits of RBPs associated with aberrant AS in HD in cortical neurons. However, we were not able to detect consistent

changes in protein insolubility of those targets in HD neurons. Only a modest, but significant increase in QKI soluble levels was observed.

We have summarized these additional analyses and observations in the results section as followed:

“Next, we asked whether the enriched RBPs exhibited a change in gene expression. Elevated levels of HLTF and a reduction of PCBP2 were found in fibroblasts, whereas a reduction of U2AF1 and U2AF2 was detected in cortical neurons of HD patients (Extended Data Figure 2A and B). A reduction of U2AF2 was previously reported in HD ¹². Protein insolubility is an important feature in neurodegenerative diseases and we have recently shown that changed biochemical solubility properties of certain RBPs are associated with aberrant AS in amyotrophic lateral sclerosis (ALS) ¹⁴. We investigated the soluble and insoluble protein fractions of Ctrl and HD cortical neurons of 5 candidate RBPs that are enriched in cortical neuron HD AS events (ILF3, QKI, U2AF2, RBFOX2 and TIAL1)

(Extended Data Figure 3A-F). ILF3, U2AF2, RBFOX2 and TIAL1 did not exhibit significant changes in the level of solubility. A modest but significant increase in soluble QKI was observed in HD cortical neurons (Extended Data Figure 3C). This suggests that aberrant RNA missplicing is present in HD patients' fibroblasts and to a larger extent in iPSC derived cortical neurons. These changes may be due to altered gene expression and biochemical properties of RBPs. In conclusion, HD patients' iPSC cortical neurons recapitulate aberrant AS as a major molecular pattern of HD”

While other observations like RBP posttranslational modifications, mislocalization, or other biochemical changes and the direct functional impact of RBPs on HD splicing (e.g. by knockout or overexpression of each of these RBPs) are highly interesting, we hope the reviewer and the editor agree that a thorough and detailed investigation of RBP dysfunction in HD may be beyond the scope of this manuscript where we focus on mechanisms of the alternative splicing modulator branaplam and reversal of aberrant alternative splicing as an important pathogenic feature in HD. We have added to the discussion that a detailed characterization of the origin of aberrant alternative splicing in HD will be necessary:

“Our findings give only a glimpse into AS in HD, but show that iPSC cortical neurons may be a powerful model to study this distinct HD associated phenotype. However, there is an urgent need for future studies thoroughly dissecting the origin of aberrant AS in HD.”

The manuscript only cites Keller et al. as targeting SH-SY5Y cells, although it also describes the effect of Branaplam in vivo in a mouse model of HD. These recently published in vivo experiments support the potential of Branaplam as a treatment for HD.

We have added the important information that Branaplam has already been administered to a mouse model of HD indicating an ameliorated disease phenotype upon Branaplam administration.

“Furthermore, the efficacy of Branaplam for HD was recently demonstrated by phenotype improvements in a HD mouse model upon administration²⁰.”

The off-target splicing effects of Branaplam were previously reported by Monteys et al (2021) Nature 596: 291-295. This reference should be cited, and whether the splicing events detected overlap with those in the present study should be addressed.

We have compared the events identified by us with the exclusive and enriched Branaplam induced events identified by Monteys et al. We find that 15/25 of the exclusive events are consistent as branaplam induced novelISS alternative splicing events. These observations obtained by orthogonal experimental and bioinformatic pipelines by us and Monteys and colleagues validate and strengthen our and their findings. We have added this analysis to Extended Data Figure 6 and described the finding in the text as followed:

“We compared our Branaplam-induced exons (cluster 6 and 9) to Branaplam-induced exons that have been previously characterized in HEK293 cells¹⁵. Interestingly, out of 25 events discovered by Monteys and colleagues to be exclusively regulated by Branaplam, 15 were also present within the novelISS exons induced by Branaplam (Extended Data Figure 5C). This suggests a very high validity and robustness of the present analyses and therefore the identified events.”

The drug reverses some aberrant splicing events associated with HD, but it induces other aberrant splicing events. Although caspase 3/7 cleavage was not induced, this is a limited way to address potential toxicity.

To further address potential toxicity, we extended our analysis. In our initial submission we provided evidence of no toxic effects of branaplam by using the adenylate kinase assay in 4 cell types and further determined the induction of apoptosis in cortical neurons. We agree that these measures do only address toxicity issues in the context of cell death but leave out other cellular systems that may be affected by application of this drug. We have therefore now added a different analyses, measuring the impact of Branaplam on proliferation in cortical progenitors. We do not observe any effect of Branaplam, confirming previous findings in animals in a human system.

“Additionally, we explored the impact of Branaplam on proliferation of SOX2⁺ neural progenitor cells via EdU incorporation assay. No changes were observed in proliferation upon 3 days Branaplam treatment (Extended Data Figure 4D and E). In summary, these findings suggest that Branaplam efficiently reduces total and mutant HTT protein levels in various Ctrl and HD patient-derived cell types without inducing toxicity and altering proliferation”

and

“This included no change human cortical progenitor proliferation, confirming previous studies that investigated proliferation in the subventricular zone of dogs and rats upon Branaplam administration ²¹.”

Reviewer #3 (Remarks to the Author):

Several approaches to lowering the levels of the huntingtin protein (HTT) are currently in clinical trials for evaluation as disease modifying treatments for Huntington's disease (HD). This includes the small molecule splicing modulator, Branaplam, which results in the inclusion of exon 49a, leading to a frame shift, and nonsense-mediated decay of the HTT transcript. In this study, the authors have investigated the effects of Branaplam in HD patient derived cells including fibroblasts and iPSC derived cortical neurons. First, they show that fibroblasts and cortical neurons recapitulate aberrant alternative splicing as a molecular fingerprint of HD. They demonstrate that Branaplam lowers total HTT and mutant HTT levels in fibroblasts, iPSCs, cortical progenitors, and neurons in a dose dependent manner at an IC50 consistently below 10nm without inducing cellular toxicity. Branaplam promotes inclusion of non-annotated novel exons, and they used an unbiased approach to identify the effects of Branaplam on splicing at the transcriptome level and the effects of this on gene expression. Branaplam was found to ameliorates aberrant alternative splicing in HD patients' fibroblasts and cortical neurons.

This is a timely study, and the effects of Branaplam on splicing throughout the transcriptome will be of great interest to those in the HD community and others interested in the actions of these splicing-modulator type drugs. Unfortunately, this paper is let down by the protein data, which cannot be presented in their current form.

We thank Reviewer 3 for the thorough review and their shared interest for alternative splicing in HD.

Introduction

Line 41: Individuals with CAG repeats of 40 and above will develop HD, not 39 and above. The authors are correct in the following sentence when they state that alleles of 36-39 exhibit incomplete penetrance.

We thank the reviewer for the careful reading and changed the sentence within the introduction accordingly.

Protein data

Figure 1E-I

These data have been interpreted incorrectly. The 2B7-MW1 assay, to all intents and purposes, is specific to mutant HTT. Therefore, the 2B7-MW1 signal in the various HD and control cell lines indicated that mutant HTT is present in the HD lines and not in the controls. It does not indicate that the level of mutant HTT was increased in the HD lines as compared to the controls. Given that the total HTT assay measures both mutant and wild type, the fact that this is constant between the HD and control lines indicates that mutant HTT and wild type HTT levels are comparable.

The MSD data are very poor given that these are supposed to be clinical-grade GLP-approved assays.

Figure 3

The data in this figure do not make sense e.g. panel D, untreated cells: The total HTT assay (control and mutant) gives a concentration of 400 pmol/g for one HD cell line. Presumably, given that Figure 1I indicated that there was no difference in total HTT levels between control and HD cells, it can be assumed that 200 pmol/g of this is comprised of wild type HTT and 200 pmol/g of mutant HTT. Yet the mutant HTT assay indicated that there are only 25 pmol/g of mutant HTT in this line.

Similarly panel E indicates that the HD and control lines each have 100 pmol/g of total HTT (HD and wild type) and yet the mutant HTT assay indicates that the level of mutant HTT alone is 100 pmol/g.

Clearly the method of protein quantification that has been used in this paper gives confusing and contradictory data. It cannot be represented in this way as it is currently meaningless.

To clarify, we used mesoscale discovery (MSD) assay-based measurements to determine changes in total (2B7/D7F7 assay) or mutant HTT (2B7/MW1 assay) protein expression separately.

The 2B7 antibody binds to the first 17 amino acids of HTT. The D7F7 antibody binds downstream of the poly-Q tract, at a single epitope in the center of HTT. In combination, 2B7 and D7F7 can be therefore used to detect HTT levels regardless of their poly-Q length (=total HTT).

The MW1 antibody is widely used to detect mutant HTT. Together with 2B7, they are currently used to detect mutant HTT in the CSF of HD patients (Wild et al., 2015 and Fondale et al., 2017) and therefore used in HTT lowering clinical trials. MW1 has a higher avidity to mHTT due to bivalent binding of the antibody at sites with elongated poly-Q. The MW1 antibody binds to an epitope of approx. 10 Qs, which means that there are several potential epitopes available within a given HTT poly-Q stretch. Thus, this poly-Q-binding antibody does not specifically, but preferentially recognize mHTT and it can be expected that the 2B7/MW1 assay will result in a much higher signal for mutant expanded HTT (McDonald et al, 2014, and Fondale et al., 2017). However, it is important to note that also non-mutant HTT with shorter poly-Q length is recognized with this antibody combination (McDonald et al, 2014, and Fondale et al., 2017). That said, at equimolar input, the signal will be significantly lower.

Unfortunately, the assay does not allow to compare the results from the total HTT assay (2B7/D7F7) and the mutant HTT assay (2B7/MW1) with each other. The numerical values from the two assays cannot be compared by mathematical addition as they are two separate assays with different antibodies that have not the exact same binding properties as explained above (e.g. D7F7 binds once, MW1 can bind multiple times). The fact that measurements of mutant and total HTT levels do not easily add up is underlined by the fact that significant effort has been made within clinical trials to develop assays that directly measure non-mutant HTT, e.g. by depleting CSF from mutant HTT and measuring residual HTT in a total HTT assay. Similar issues are encountered in other poly-Q protein assays, e.g. for the detection of ATXN3, where total ATXN3 and mutant ATXN3 levels (the latter determined also using the MW1 antibody) will seemingly deviate. Thus, these two assays should be seen as a semi-quantitative approach to visualize the changes in raw signal and give an indication to the relative changes in protein levels, rather than absolute levels. We acknowledge the reviewers concerns about potential confusions and have added in the figure legend, methods and results section a paragraph explaining the interpretation of the obtained values:

In figure legend 1E:

“Numeric values from 2B7/D7F7 assay (total HTT) cannot be directly set in relation to numeric values from 2B7/MW1 assay (mutant HTT).”

In the results part:

“Next, we determined the total (tHTT) and mHTT levels in cellular homogenates. We used a meso scale discovery (MSD) assay with an electro-chemiluminescent readout. MSD assays are applied to semi-quantitatively determine changes in protein levels. An N-terminally-binding HTT antibody (2B7) captures HTT and an antibody binding a central part of HTT downstream of the poly-Q tract (D7F7) or an antibody with preferred binding to elongated poly-Q (MW1) is used to quantify tHTT (2B7/D7F7) and mHTT (2B7/MW1) via a SULFO-TAG, respectively. The resultant HTT signal values are back-calculated to a standard of recombinant HTT with 23 (Q23) and 73 (Q73) glutamines (Figure 1E). Importantly, the obtained signal values of the tHTT assay (2B7 and D7F7) and mHTT assay (2B7 and MW1) in a given sample cannot be directly set into relation with each other (e.g. no mHTT/tHTT ratio calculations possible) due to different properties of both assays (explained in detail in methods section)”

In the methods section:

“A first antibody (2B7) is used to capture HTT and a second antibody (D7F7 or MW1) is used to detect and quantify HTT via a SULFO-TAG. The 2B7 antibody binds to the first 17 amino acids of HTT. The D7F7 antibody binds downstream of the poly-Q tract, at a single epitope in the center of HTT. In combination, 2B7 and D7F7 can be therefore used to detect total HTT levels regardless of their poly-Q length (tHTT). The MW1 antibody is widely used to detect mutant HTT. Together with 2B7, they are currently used to detect mutant HTT in the CSF of HD patients^{21,22} and therefore used in HTT lowering clinical trials. MW1 has a higher avidity to mHTT due to bivalent binding of the antibody at sites with elongated poly-Q. Thus, this poly-Q-binding antibody do not specifically, but preferentially recognize mHTT and it can be expected that the 2B7/MW1 assay will result in a higher signal for mutant, expanded HTT^{22,23}. However, it is important to note that also non-mutant HTT with shorter poly-Q length is recognized with this antibody combination^{22,23}. ”

and

“The signal values were back-calculated to the standard that was run in parallel (Q23 for 2B7/D7F7 assay and Q73 for 2B7/MW1 assay), resulting in MSD signal values of a sample equivalent to a certain HTT Q23 or HTT Q73 concentration.

The assay does not allow to compare the numerical results from the total HTT assay and the mutant HTT assay with each other. The numerical values from both assays cannot be compared by mathematical addition as they are two separate assays with different antibodies that have not the identical binding properties (e.g. D7F7 binds once, MW1 can bind multiple times depending on poly-Q length).”

To allow a comparison of the obtained signal values of a given assay and individual between different experiments (e.g. mHTT quantification of a donor 1 in fibroblasts vs. donor 1 in iPSC), the MSD intensity values are back-calculated using a standard of a recombinant protein containing for the 2B7/D7F7 assay a Q-length of 23 (Q23) and for the 2B7/MW1 assay a Q-length of 73 (Q73). Since individuals have a varying degree of poly-Q lengths that will introduce differences in kinetics and avidity of the assay, the resultant values have to be interpreted as signal equivalent to signal resulted from a molar concentration of HTT peptide with Q23 (for 2B7/D7F7) or Q73 (2B7/MW1). We have clarified this in the methods section and changed the y-axes labels for all MSD results to the following:

‘MSD signal equivalent to Q23 [pmol/g]’

‘MSD signal equivalent to Q73 [pmol/g]’

We hope that these clarifications help to resolve potential confusions of the MSD assay results and enhance the proper understanding and interpretation of our results for the reader.

References

- Fodale, Valentina et al. ‘Validation of Ultrasensitive Mutant Huntingtin Detection in Human Cerebrospinal Fluid by Single Molecule Counting Immunoassay’. 1 Jan. 2017 : 349 – 361.
- Wild EJ, Boggio R, Langbehn D, Robertson N, Haider S, Miller JR, Zetterberg H, Leavitt BR, Kuhn R, Tabrizi SJ, Macdonald D, Weiss A. Quantification of mutant huntingtin protein in cerebrospinal fluid from Huntington's disease patients. *J Clin Invest.* 2015 May;125(5):1979-86. doi: 10.1172/JCI80743. Epub 2015 Apr 6. PMID: 25844897; PMCID: PMC4463213.
- Macdonald D, Tessari MA, Boogaard I, Smith M, Pulli K, Szyndol A, Albertus F, Lamers MB, Dijkstra S, Kordt D, Reindl W, Herrmann F, McAllister G, Fischer DF, Munoz-Sanjuan I. Quantification assays for total and polyglutamine-expanded huntingtin proteins. *PLoS One.* 2014 May 9;9(5):e96854. doi: 10.1371/journal.pone.0096854. PMID: 24816435; PMCID: PMC4016121.

REVIEWERS' COMMENTS

Reviewer #1 (Remarks to the Author):

Thank you to the authors for the additional information and improvements to the manuscript.

Reviewer #2 (Remarks to the Author):

I'm sorry for the delay in reviewing this resubmission.

I feel that the authors have made a reasonable effort to address the points I raised in my previous review. Overall, the manuscript is not especially novel, in light of the Keller et al reference in Nat Comm, but it provides an incremental advance in an important area, so it may be appropriate for publication in Nat Comm.

My only additional suggestion at this stage would be to provide some further clarification in the Discussion regarding the following two statements: "Our iPSC-based model recapitulates aberrant alternative splicing as a feature of HD that has been previously observed in postmortem tissue"; and "In conclusion, HD patients' iPSC cortical neurons recapitulate aberrant AS as a major molecular pattern of HD".

The profile of differential splicing does not seem to be similar, at least for the candidate genes in ref. 12 (CCDC88C, KCTD17, SYNJ1, VPS13C, TRPM7 and SLC9A5). Aberrant alternative splicing should probably not be considered to be a "pattern" in common, because the AS profile seems to be totally different between their cortical-neuron model and post-mortem brain, even if this is because of differences in models or disease stages. The difference in these AS patterns needs to be mentioned to avoid overinterpretation that "Branaplam treatment can reverse the AS phenotype happening in HD brains". The AS in ref. 12 and the AS they observed are both "aberrant AS", but they are not the same.

As the reversal of aberrant AS is an important feature of Branaplam treatment unique to this manuscript, it should be made even clearer that it is still not known if this treatment is suppressing specific AS events that cause the disease.

Reviewer #3 (Remarks to the Author):

The authors have satisfied the concerns of this referee.

Response to reviewer comments

Answers are indented.

Reviewer #1 (Remarks to the Author):

Thank you to the authors for the additional information and comments to the manuscript.

We thank Reviewer #1 for acknowledging our improvements and for his/her support for the manuscript.

Reviewer #2 (Remarks to the Author):

I'm sorry for the delay in reviewing this resubmission.

I feel that the authors have made a reasonable effort to address the points I raised in my previous review. Overall, the manuscript is not especially novel, in light of the Keller et al reference in Nat Comm, but it provides an incremental advance in an important area, so it may be appropriate for publication in Nat Comm.

We thank Reviewer #2 for acknowledging our efforts and her/his approval of the manuscript.

My only additional suggestion at this stage would be to provide some further clarification in the Discussion regarding the following two statements: "Our iPSC-based model recapitulates aberrant alternative splicing as a feature of HD that has been previously observed in postmortem tissue"; and "In conclusion, HD patients' iPSC cortical neurons recapitulate aberrant AS as a major molecular pattern of HD".

The profile of differential splicing does not seem to be similar, at least for the candidate genes in ref. 12 (CCDC88C, KCTD17, SYNJ1, VPS13C, TRPM7 and SLC9A5). Aberrant alternative splicing should probably not be considered to be a "pattern" in common, because the AS profile seems to be totally different between their cortical-neuron model and post-mortem brain, even if this is because of differences in models or disease stages. The difference in these AS patterns needs to be mentioned to avoid overinterpretation that "Branaplam treatment can reverse the AS phenotype happening in HD brains". The AS in ref. 12 and the AS they observed are both

“aberrant AS”, but they are not the same.

As the reversal of aberrant AS is an important feature of Branaplam treatment unique to this manuscript, it should be made even clearer that it is still not known if this treatment is suppressing specific AS events that cause the disease.

To clarify, it was not our intention to claim that the identical AS events occur in postmortem tissue and iPSC-neurons. Considering this important point, we will avoid to state that ‘Branaplam treatment can reverse the AS phenotype happening in HD brains’. We adapted accordingly by specifying the mentioned statements. Despite the fact, that the candidate AS events are different to published post-mortem studies, there is an overlap of the RBPs (e.g. U2AF2, RBFOX2 and TIA1) driving AS observed in postmortem tissue and iPSC-neurons. This suggests, that similar alterations in RNA metabolism may occur in postmortem and the present in vitro model. We have added more details and further clarifications to emphasize these specifications:

In the results:

‘This suggests that Branaplam ameliorates a prominent molecular signature **in HD iPSC-derived cortical neurons.**’

In the discussion:

‘We further underscore the effectiveness of this molecule by providing compelling evidence that Branaplam ameliorates a molecular fingerprint in a human in vitro HD model. However, it is of speculative nature if rescue of these AS events contributes to the clinical efficacy of the drug.’

and

‘Our iPSC-based model exhibits aberrant alternative splicing, a feature also present in HD postmortem tissue ^{11,12}. Six proposed candidate AS events in postmortem tissue (within the transcripts of CCDC88C, KCTD17, SYNJ1, VPS13C, TRPM7, SLC9A5) are not recapitulated in our iPSC-based neuronal dataset. However, there appears to be a similarity of the RNA-binding proteins driving aberrant AS in the present iPSC neuronal and previously published

postmortem tissue¹², suggesting the possibility of shared changes in RNA-binding protein function and RNA processing .'

Reviewer #3 (Remarks to the Author):

The authors have satisfied the concerns of this referee.

We thank Reviewer #3 for his/her approval of the manuscript.